# Invariance Makes LLM Unlearning Resilient Even to Unanticipated Downstream Fine-Tuning

**Changsheng Wang** [1]  **Yihua Zhang** [1]  **Jinghan Jia** [1]  **Parikshit Ram** [2]  **Dennis Wei** [2]  **Yuguang Yao** [1]
**Soumyadeep Pal** [1]  **Nathalie Baracaldo** [2]  **Sijia Liu** [1 2]

## Abstract

Machine unlearning offers a promising solution to privacy and safety concerns in large language models (LLMs) by selectively removing targeted knowledge while preserving utility. However, current methods are highly sensitive to downstream fine-tuning, which can quickly recover forgotten information—even from unrelated tasks. To address this, we introduce *invariance* into unlearning for the first time, inspired by invariant risk minimization (IRM). Building on this principle, we propose invariant LLM unlearning (ILU), a regularization-based framework that enhances robustness. Notably, ILU generalizes well to diverse fine-tuning tasks, even when trained using a single dataset. A task vector analysis is also provided to further elucidate the rationale behind ILU's effectiveness. Extensive experiments on the WMDP and MUSE benchmark, reveal that ILU significantly outperforms state-of-the-art unlearning methods, including negative preference optimization (NPO) and representation misdirection for unlearning (RMU). Notably, ILU achieves superior unlearning robustness across diverse downstream fine-tuning scenarios (*e.g.*, math, paraphrase detection, and sentiment analysis) while preserving the fine-tuning performance. Our experiments and codes are available at https://github.com/OPTML-Group/Unlearn-ILU.

## 1. Introduction

Large language models (LLMs) have revolutionized generative AI (Touvron et al., 2023; Achiam et al., 2023; Liu et al., 2024a). However, their extensive training on diverse corpora introduces critical ethical and security risks, including privacy violations through memorization of sensitive data (Huang et al., 2024; Shi et al., 2024), amplification of societal biases (Motoki et al., 2023), and generation of harmful or illegal content (Wen et al., 2023; Li et al., 2024). These challenges necessitate effective mechanisms to eliminate undesirable data-model influences in pre-trained models without compromising their utility—a problem referred to as **LLM unlearning** (Liu et al., 2024c; Maini et al., 2024; Yao et al., 2024b).

Existing LLM unlearning methods often operate under the assumption of being *standalone* interventions for safe model deployment (Yao et al., 2024a; Zhang et al., 2024a; Li et al., 2024; Gao et al., 2024; Cooper et al., 2024; Liu et al., 2024b), meaning they do not account for subsequent operations post-unlearning. However, in practice, unlearning is rarely the final optimization step, as industrial pipelines often apply subsequent fine-tuning to adapt models for downstream tasks. Recent empirical studies (Barez et al., 2025; Łucki et al., 2024; Hu et al., 2024; Tamirisa et al., 2024; Lynch et al., 2024) reveal a critical vulnerability in current unlearning approaches: knowledge erased during unlearning can unexpectedly *resurface* through downstream fine-tuning, even on data unrelated to the unlearning objective.

This vulnerability suggests that existing unlearning approaches only remove unwanted knowledge *temporarily*. However, unlearning should ideally be permanent for a large variety of use cases including removing copyrighted or harmful material. Therefore, we ask:

*Q: How can we define 'invariance' and integrate it into LLM unlearning to enhance resilience against fine-tuning?*

Addressing **Q** necessitates a fundamental rethinking of LLM unlearning optimization to account for the impact of *future* model adaptations, a highly non-trivial challenge given the *unforeseen* nature of downstream fine-tuning operations. One straightforward yet computationally intensive approach is to leverage meta-learning (Finn et al., 2017; Tamirisa et al., 2024), which frames the unlearned model as a meta-model designed to be agnostic to downstream fine-tuning. However, this approach could be challenging to scale due

---

[1]Michigan State University [2] IBM Research. Correspondence to: Changsheng Wang <wangc168@msu.edu>, Sijia Liu <liusiji5@msu.edu>.

*Proceedings of the $42^{st}$ International Conference on Machine Learning*, Vancouver, Canada. PMLR 267, 2025. Copyright 2025 by the author(s).

to the need for gradient unrolling to simulate fine-tuning paths during the optimization process (Tamirisa et al., 2024; Zhang et al., 2024b).

Thus, instead of meta-learning, we explore invariance in the unlearned model—ensuring that forgotten knowledge remains irretrievable through parameter updates from irrelevant downstream fine-tuning tasks. To model and promote 'invariance', we adopt the technique of invariant risk minimization (IRM) (Arjovsky et al., 2019), incorporating its invariance regularization (which achieves invariant predictions agnostic to different training environments) into LLM unlearning. If downstream fine-tuning is treated as an environment that challenges unlearning, invariance regularization can be then integrated with the unlearning objective to obtain unlearning resilience against fine-tuning.

In our work, the integration of IRM into unlearning leads to a new regularized optimization framework, termed invariant LLM unlearning (**ILU**), which optimizes the model to remain stationary under fine-tuning perturbations during unlearning, aiming to "immunize" against the revival of erased knowledge through fine-tuning. In addition, ILU employs lightweight gradient regularization that integrates seamlessly with state-of-the-art (SOTA) unlearning methods like negative preference optimization (NPO) (Zhang et al., 2024a) and representation misdirection for unlearning (RMU) (Li et al., 2024), avoiding the computational overhead of meta-learning while enabling robust knowledge erasure when subjected to fine-tuning. Furthermore, ILU generalizes effectively to diverse fine-tuning datasets, including those absent during (invariant) training.

We summarize **our contributions** below:

❶ We introduce the concept of invariance into LLM unlearning, establishing a novel connection between IRM and LLM unlearning to enhance resilience against downstream fine-tuning. This integration results in ILU.

❷ We demonstrate that the invariance regularization in ILU, even with a simple formulation built upon a single unrelated fine-tuning set, can effectively enhance the resilience of an unlearned model against new, unforeseen downstream fine-tuning tasks. We also conduct a task vector analysis to elucidate the underlying rationale behind ILU.

❸ We show that ILU enhances SOTA LLM unlearning methods, such as NPO and RMU, when subjected to downstream fine-tuning. For instance, on the WMDP unlearning benchmark, ILU achieves a 23% average robustness improvement over RMU across 6 fine-tuning tasks, while maintaining fine-tuning accuracy.

## 2. Related Work

**Machine unlearning in LLMs.** Recent advances in machine unlearning for LLMs have shown promise in addressing risks associated with undesired data retention (Liu et al., 2024c; Yao et al., 2024a; Zhuang et al., 2024; Maini et al., 2024; Eldan & Russinovich, 2023). Practical implementations span critical applications, such as privacy protection through the removal of sensitive information (Wu et al., 2023; Yu et al., 2023), prevention of harmful content generation (Lu et al., 2022; Li et al., 2024), and elimination of memorized sequences (Barbulescu & Triantafillou, 2024; Jang et al., 2023). Most LLM unlearning methods rely on effective and efficient optimization techniques to avoid computationally prohibitive retraining while aiming to 'faithfully' remove unwanted data-model influences (Liu et al., 2024c). For instance, regularized optimization (Yao et al., 2024b; Liu et al., 2024c; Li et al., 2024; Zhang et al., 2024a) has been predominantly employed to balance unlearning effectiveness with preserved model utility post-unlearning. Some approaches employ localized interventions that target specific model components associated with unwanted capabilities (Meng et al., 2022; Wei et al., 2024; Jia et al., 2024). Other unlearning approaches leverage in-context learning (Pawelczyk et al., 2023; Thaker et al., 2024) or task vector (Ilharco et al., 2023) to negate the effects of unwanted data or model capabilities in LLMs.

**Robustness challenge in LLM unlearning.** Despite the growing importance of LLM unlearning, recent studies have revealed significant shortcomings in the robustness of current unlearning methods (Deeb & Roger, 2024). For example, Lynch et al. (2024) demonstrated that standardized evaluation protocols often overlook residual knowledge persisting in unlearned models. This empirical finding is consistent with adversarial analyses in Łucki et al. (2024), where fine-tuning on just 10 unrelated examples restores most of the unlearned information. The so-called relearning attacks (Hu et al., 2024) further underscore the robustness limitations of LLM unlearning, showing that a small amount of unlearning-related data can quickly recover most of the unlearned information through fine-tuning. The above vulnerabilities align with a broader limitation of LLMs: weight modifications (such as fine-tuning) can undermine previously applied alignment operations (Qi et al., 2024; Jain et al., 2024). Indeed, additional evidence includes erased concepts resurfacing through neuron repurposing (Lo et al., 2024) and quantization attacks restoring unlearned knowledge (Zhang et al., 2024c). Compared to 'attacking' unlearned models, efforts to enhance the robustness of LLM unlearning are quite limited. One concurrent study (Fan et al., 2025) adopts smoothness-oriented optimization techniques, *e.g.*, sharpness-aware minimization (SAM), to improve resistance to relearning by promoting local flatness

in the forget loss landscape. Latent adversarial training (Sheshadri et al., 2024) strengthens robustness by perturbing intermediate activations, thereby suppressing undesirable behaviors and reducing relearning potential. Tamper-resistant safeguards (Tamirisa et al., 2024) show promise but involve meta-learning-like complex optimization. Additionally, recent findings (Barez et al., 2025) reveal that existing unlearning algorithms struggle in dual-use knowledge scenarios, where safety and utility objectives conflict. Unlike existing approaches, our work promotes invariance in LLM unlearning to ensure robustness against post-fine-tuning operations.

**Invariant risk minimization.** The conceptual foundation of invariant learning originates from causal inference frameworks (Peters et al., 2016), with Arjovsky et al. (2019) pioneering its adaptation for machine learning and practical adoption. Subsequent advances have expanded IRM's technical landscape across various dimensions, such as risk variance regularization across domains (Krueger et al., 2021; Xie et al., 2020), gradient distribution alignment (Rame et al., 2022), and adversarial domain adaptation via regret minimization (Jin et al., 2020). Other innovations include Bayesian uncertainty quantification (Lin et al., 2022), sparse feature selection mechanisms (Zhou et al., 2022), bi-level optimization (Zhang et al., 2023), second-order optimization (Zhang et al., 2023), and ensemble-based environment specialization (Ahuja et al., 2020). In this work, we establish a connection between IRM and unlearning to enhance the latter's resilience against downstream fine-tuning.

## 3. Preliminaries and Problem Statement

**LLM unlearning.** Achieving the complete removal of undesired data-model influences from an LLM–without compromising model utility and without requiring full model retraining–remains a significant challenge for LLM unlearning (Liu et al., 2024c). This necessitates the careful design of a *forget* objective function to enforce unlearning, regularized by a (utility-driven) *retain* objective to balance unlearning efficacy and model utility (Zhang et al., 2024a; Li et al., 2024; Maini et al., 2024). Accordingly, the problem formulation of LLM unlearning can be cast as:

$$\underset{\boldsymbol{\theta}}{\text{minimize}} \quad \ell_{\mathrm{u}}(\boldsymbol{\theta}; \mathcal{D}_{\mathrm{f}}, \mathcal{D}_{\mathrm{r}}) := \ell_{\mathrm{f}}(\boldsymbol{\theta}; \mathcal{D}_{\mathrm{f}}) + \gamma \ell_{\mathrm{r}}(\boldsymbol{\theta}; \mathcal{D}_{\mathrm{r}}), \quad (1)$$

where $\boldsymbol{\theta}$ denotes the model parameters to be optimized from a pre-trained state. The unlearning objective, $\ell_{\mathrm{u}}$, comprises the forget objective, $\ell_{\mathrm{f}}$, which is defined over the forget set $\mathcal{D}_{\mathrm{f}}$, and the retain objective, $\ell_{\mathrm{r}}$, which regularizes model utility using the retain set $\mathcal{D}_{\mathrm{r}}$. The parameter $\gamma \geq 0$ serves as a regularization factor to balance forget and retain objectives.

In practice, the retain objective $\ell_{\mathrm{r}}$ is often chosen as the cross-entropy-based sequence prediction loss, aligning with

the original training objective. However, the design of the forget objective $\ell_{\mathrm{f}}$ requires more careful consideration, with two widely adopted approaches. The first approach, known as negative preference optimization (NPO) (Zhang et al., 2024a), treats the forget data in $\mathcal{D}_{\mathrm{f}}$ as negative samples, effectively disrupting the model's retention of these negative samples. The second approach leverages random features, aligning the representations of the forget data with random vectors to enforce unlearning, which is referred to as representation misdirection for unlearning (RMU) (Li et al., 2024). We refer readers to the corresponding literature for detailed formulations of NPO and RMU.

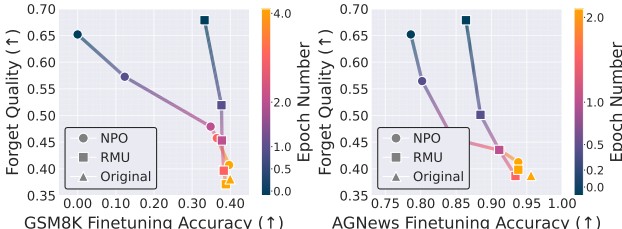

*Figure 1.* Fine-tuning breaks existing unlearning methods. Performance evaluation of popular unlearning methods, NPO and RMU, applied to the LLM Zephyr-7b-beta for removing harmful knowledge generation on the WMDP dataset (Li et al., 2024). The effectiveness of unlearning is measured by the accuracy of the unlearned model on the WMDP-Bio evaluation set, with lower accuracy indicating better forgetting. Accordingly, we define 'forget quality' as '1 - evaluation accuracy', where a higher value means more effective unlearning. **(Left: GSM8K fine-tuning)** The trajectory of forget quality and fine-tuning accuracy is presented for various models, including NPO or RMU-unlearned models and the original (non-unlearned) model, when subjected to downstream fine-tuning on the GSM8K dataset. The fine-tuning epoch number is indicated by the color, ranging from 0 (no fine-tuning) to the number required to achieve lossless performance equivalent to full fine-tuning of the original model (termed 'Original'). The dots with the same position (*e.g.*, 1st, 2nd, 3rd) and color across NPO's and RMU's trajectories represent the same fine-tuning epoch number. **(Right: AGNews fine-tuning)** Similar to the left plots but applied to fine-tuning on the AGNews downstream dataset.

**Unlearning vulnerability to downstream fine-tuning.** Recent studies (Barez et al., 2025; Łucki et al., 2024; Hu et al., 2024; Tamirisa et al., 2024; Lynch et al., 2024) have empirically demonstrated that knowledge removed through unlearning optimization (1) can be rapidly recovered via post-unlearning fine-tuning, even when fine-tuning is performed using data entirely unrelated to the unlearning. We refer to this unlearning challenge as the *vulnerability of an unlearned model to downstream fine-tuning*.

In **Fig. 1**, we demonstrate the unlearning effectiveness and fine-tuning performance of unlearned models (obtained by NPO and RMU) against a varying number of fine-tuning epochs on two downstream datasets: GSM8K and AGNews. Here unlearning is applied to the model Zephyr-7b-beta on

the WMDP dataset (Li et al., 2024) for harmful content degeneration and evaluated using *1 minus accuracy on the WMDP-Bio evaluation set*, termed as 'forget quality', where a higher value indicates better unlearning performance. For comparison, we also include the performance of a model fine-tuned from the original pre-trained state (*i.e.*, 'Original'). As we can see, both NPO and RMU lose their unlearning effectiveness during the fine-tuning process, despite (1) having high forget quality prior to fine-tuning (*i.e.*, at 0 fine-tuning epochs) and (2) the downstream tasks (GSM8K or AGNews) being unrelated to the unlearning task (WMDP). In addition, as the number of fine-tuning epochs increases, the fine-tuned models (even when starting from different initial unlearned models) exhibit improved fine-tuning accuracy, converging to the performance of fine-tuning the 'Original' model. Furthermore, while RMU yields a higher forget quality prior to fine-tuning, it demonstrates weaker robustness to downstream fine-tuning, as evidenced by the faster *un*unlearning rate with increasing fine-tuning epochs.

**Problem statement.** As motivated by Fig. 1, fine-tuning on unrelated information that reverses unlearning has emerged as a significant concern and a major limitation of current LLM unlearning methods. Thus, the central problem addressed in this paper is: *How can we enhance the current LLM unlearning approach (1) to achieve greater resilience against downstream fine-tuning?*

We will approach the above problem through the lens of **IRM** (invariant risk minimization). By leveraging IRM, we can integrate and promote invariance in LLM unlearning.

## 4. Promoting Invariance in LLM Unlearning

**Invariance in LLM unlearning: From IRM to ILU.** IRM is designed to seek an "invariant" (*i.e.*, training environment-agnostic) model, aiming for universal representation learning to achieve optimal predictions across diverse training environments (Arjovsky et al., 2019; Ahuja et al., 2020; Zhang et al., 2023). By treating a fine-tuning task as an unlearning training environment, integrating IRM with (1) is then expected to enhance the invariance of the unlearned model, *i.e.*, improve its robustness against fine-tuning. Therefore, we propose framing the LLM unlearning problem within the IRM framework.

Following the IRM setup in (Arjovsky et al., 2019), let $\mathcal{D}_i$ denote the dataset associated with the training environment $i \in [N]$, where $[N] := \{1, 2, \dots, N\}$, and $N$ represents the number of training environments. IRM learns a universal representation model $\phi$, which enables the existence of an invariant predictor $\mathbf{w}$ that remains simultaneously optimal across all environments. This results in the invariant (training environment-agnostic) model $\boldsymbol{\theta} := \mathbf{w} \circ \phi$ (where $\circ$ denotes model composition), which not only optimizes the

empirical training objective for the representation model $\phi$ but also ensures optimal fine-tuning performance on each dataset $\mathcal{D}_i$. Formally, the IRM problem is formulated as

$$
\begin{aligned}
\underset{\phi}{\text{minimize}} \quad & \ell_{\text{ERM}}(\mathbf{w}^*(\phi) \circ \phi; \cup_i \mathcal{D}_i) \\
\text{subject to} \quad & \mathbf{w}^*(\phi) \in \arg\min_{\mathbf{w}} \ell_i(\mathbf{w} \circ \phi; \mathcal{D}_i), \ \ \forall i \in [N],
\end{aligned} \tag{2}
$$

where the upper-level objective $\ell_{\text{ERM}}(\cdot; \cup_i \mathcal{D}_i)$ represents empirical risk minimization (ERM) across all data to optimize $\phi$ (*e.g.*, $\ell_{\text{ERM}}(\cdot) = \sum_i \ell_i(\cdot; \mathcal{D}_i)$), $\ell_i$ denotes the individual training loss over $\mathcal{D}_i$, and $\mathbf{w}^*(\phi)$ denotes an invariant predictor (built upon $\phi$) that is optimal to any specific training environment. Here we explicitly express the solution $\mathbf{w}^*(\phi)$ as a function of $\phi$.

However, solving the original IRM problem (2) poses significant challenges due to the need to compute the gradient $\frac{d\mathbf{w}^*(\phi)}{d\phi}$. To circumvent that, problem (2) is typically relaxed to a single-level, regularized problem, referred to as IRMv1:

$$
\underset{\boldsymbol{\theta}}{\text{minimize}} \ \underbrace{\ell_{\text{ERM}}(\boldsymbol{\theta})}_{\text{ERM}} + \lambda \underbrace{\sum_{i=1}^{N} \| \nabla_w \ell_i(w \circ \boldsymbol{\theta}; \mathcal{D}_i) \,|_{w=1} \|_2^2}_{\text{Invariance regularization}} \tag{3}
$$

where the original predictor's parameters $\mathbf{w}$ are absorbed into the full model parameters $\boldsymbol{\theta}$, $\lambda > 0$ serves as a regularization parameter, $\| \cdot \|_2$ is the $\ell_2$ norm, and $\nabla_w \ell_i$ represents the gradient of $\ell_i$ with respect to the (virtual) scalar predictor $w$, evaluated at $w = 1$. The invariance regularization in (3) enforces the necessary optimality condition for each lower-level problem in (2) by penalizing non-stationarity.

The IRMv1 approach (3), while a relaxation of (2), facilitates the application of invariance regularization to LLM unlearning. First, composing $\boldsymbol{\theta}$ with a constant scalar predictor $w = 1$ ensures adaptability to a wide range of machine learning models, including LLMs. Second, if $\boldsymbol{\theta}$ represents the unlearned model obtained by replacing ERM with the unlearning objective $\ell_u$ in (1) and $\mathcal{D}_i$ is a fine-tuning dataset, the invariance regularization in (3) enforces robustness of $\boldsymbol{\theta}$ against fine-tuning. Therefore, we propose **ILU** (invariant LLM unlearning), extended from IRMv1 (3), as below:

$$
\underset{\boldsymbol{\theta}}{\text{minimize}} \ \ell_u(\boldsymbol{\theta}) + \lambda \sum_{i=1}^{N} \| \nabla_{w|w=1} \ell_i(w \circ \boldsymbol{\theta}; \mathcal{D}_i) \|_2^2, \tag{4}
$$

where $\ell_u$ is the unlearning objective, and $\mathcal{D}_i$ is a fine-tuning dataset that can be unrelated to the unlearning task, *e.g.*, GSM8K or AGNews in Fig. 1.

**A single (unrelated) fine-tuning dataset may suffice to promote unlearning invariance.** A key question in (4) is whether the introduction of *multiple* fine-tuning sets $\{\mathcal{D}_i\}$ (*i.e.*, $N > 1$) is necessary, as it assumes greater access to additional data and knowledge of potential fine-tuning tasks. An ideal ILU framework should minimize reliance

on fine-tuning datasets while demonstrating generalization to *unseen* ones at test time. To explore this, we focus on utilizing only a single fine-tuning dataset ($\mathcal{D}$) in (4):

$$\underset{\boldsymbol{\theta}}{\text{minimize}} \ \ell_{\mathrm{u}}(\boldsymbol{\theta}) + \lambda\|\nabla_{w|w=1}\ell_i(w \circ \boldsymbol{\theta}; \mathcal{D})\|_2^2. \qquad (5)$$

We then consider two specifications for $\mathcal{D}$ depending on the relationship with the unlearning task ($\mathcal{D}_{\mathrm{f}}$): **(a)** $\mathcal{D} \perp \mathcal{D}_{\mathrm{f}}$, indicating that the fine-tuning set is unrelated to the forget set. Here, $\perp$ denotes (nearly) zero cosine similarity between the corresponding task vectors, *e.g.*, GSM8K vs. WMDP in Fig. 1; **(b)** $\mathcal{D} = \mathcal{D}_{\mathrm{f}}$, where the forget set itself is used as the fine-tuning set during invariance regularization. The latter case is inspired by a specific fine-tuning scenario, known as *relearning attack* (Hu et al., 2024), where the forget data samples are used to fine-tune the unlearned model. The speed of relearning the forgotten knowledge is then evaluated as a measure of unlearning effectiveness.

For both ILU settings (a) and (b), we investigate the effectiveness of (5) in alleviating unlearning vulnerability against downstream fine-tuning at test time when *new* datasets $\mathcal{D}' \neq \mathcal{D}$ are used. Extending from the experiments in Fig. 1, **Fig. 2** illustrates the performance of ILU, achieved by solving (5) with the RMU-based unlearning objective for the WMDP unlearning task. We term **ILU($\mathcal{D}$)** as the

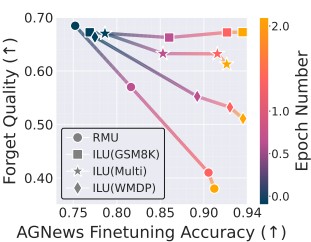

*Figure 2.* A single fine-tuning dataset suffices for enhancing unlearning robustness. Forget quality and fine-tuning accuracy of different unlearned models are presented against AGNews fine-tuning, following a similar setup and presentation format to Fig. 1.

ILU variant that adopts $\mathcal{D}$-based invariance regularization. In case (a) $\mathcal{D} \perp \mathcal{D}_{\mathrm{f}}$, ILU($\mathcal{D}$) is implemented as ILU(GSM8K). In case (b) $\mathcal{D} = \mathcal{D}_{\mathrm{f}}$, it corresponds to ILU(WMDP). For comparison, we also consider ILU based on multiple fine-tuning datasets (4), termed as **ILU(Multi)**, using GSM8K, AGNews, and WinoGrande as the fine-tuning datasets.

As shown in Fig. 2, all ILU variants demonstrate greater robustness than RMU as the fine-tuning epoch number increases. In particular, ILU(GSM8K), which uses only a single fine-tuning set $\mathcal{D}$ (unrelated to both the forget set $\mathcal{D}_{\mathrm{f}}$ and the evaluation fine-tuning set $\mathcal{D}'$), achieves the highest robustness and maintains it consistently across fine-tuning epochs, even as the model approaches full fine-tuning performance. Additionally, ILU(WMDP) underperforms compared to ILU(GSM8K), resulting in weaker robustness against downstream fine-tuning. This is unsurprising, as invariance in unlearning is conceptually better achieved using an unlearning-unrelated dataset for invariance regularization

in (5). Otherwise, a conflict arises between the unlearning objective (which aims to lower accuracy on the forget set) and forget set-based invariance regularization (which may increase accuracy on the forget set to satisfy stationarity). Furthermore, although ILU(Multi) incorporates multiple fine-tuning sets (GSM8K, AGNews, and WinoGrande) for invariance regularization in (4), it does not exhibit a robustness advantage over the simpler ILU(GSM8K) approach. This might be because incorporating more fine-tuning sets introduces additional optimization complexities: The unlearning direction needs to align with multiple fine-tuning directions and cannot contradict any of them. Based on the above, *unless stated otherwise, we implement ILU (5) with $\mathcal{D} \perp \mathcal{D}_{\mathrm{f}}$ in the rest of the paper*. We also refer readers to **Appendix B.1** for more details on the comparison between NPO and its ILU enhancement.

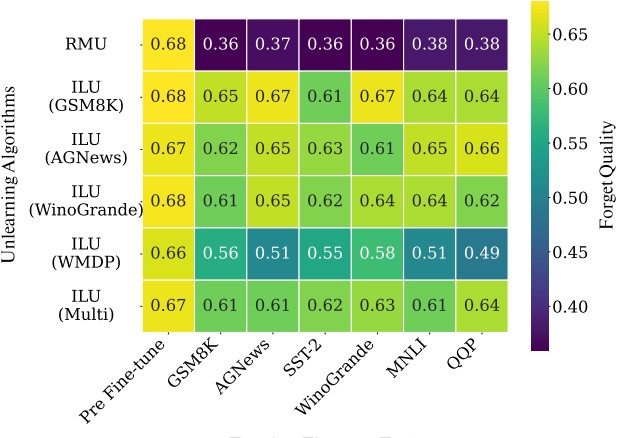

*Figure 3.* Graceful generalization of ILU's robustness to unseen fine-tuning tasks during evaluation. Heatmap of forget quality on WMDP is presented for RMU and its ILU variants to demonstrate unlearning robustness under various unlearning training and downstream fine-tuning settings, where the unlearning setup is consistent with Fig. 2, and the forget quality in each cell is reported at the final fine-tuning epoch. Each row corresponds to an unlearning training approach, while each column represents an evaluation setting (*e.g.*, a fine-tuning dataset or no fine-tuning).

As an extension of Fig. 2, **Fig. 3** compares the WMDP unlearning performance of various ILU variants with RMU, evaluated across different downstream fine-tuning datasets. See additional results on NPO in **Appendix B.2**. As observed, ILU($\mathcal{D}$), when realized with a single unlearning-irrelevant fine-tuning set (*i.e.*, $\mathcal{D}$ is either GSM8K, AGNews, or WinoGrande), enables the unlearned model to be resilient against even unseen downstream fine-tuning evaluations. The resulting forget quality not only outperforms RMU but also demonstrates advantages over ILU(WMDP) and ILU(Multi). This aligns with the insights drawn from Fig. 2. The cross-assessment in Fig. 3 (different training and testing datasets) demonstrates that incorporating invariance into LLM unlearning can strongly generalize

its robustness to unseen fine-tuning tasks (which differ from those used in invariance regularization).

## 5. Understanding ILU via Task Vectors

To understand the effectiveness of ILU in building resilience against fine-tuning, we examine the relationship between the 'unlearning direction' and 'fine-tuning direction' using task vector analysis (Ilharco et al., 2023).

A task vector defines a direction in the weight space for a specific task, where movement along this direction from the pre-trained model enhances performance on this task. Let $\theta_u$ and $\theta_o$ denote the unlearned model (obtained via an unlearning approach) and the original pre-trained model, respectively. By the definition of task vector, the **unlearning direction** (*i.e.*, unlearning task vector) is given by $\tau_u = \theta_u - \theta_o$. Specifically, $\tau_{ILU}$ (or $\tau_{NPO}$) represents the unlearning direction resulting from ILU (or NPO), respectively. Similarly, we can define the **fine-tuning direction** based on $\theta_o$ by $\tau_{ft} = \theta_{ft} - \theta_o$, where $\theta_{ft}$ is the fine-tuned model from $\theta_o$. Note that the fine-tuning direction resides in the *un*unlearning space, as fine-tuning alone cannot achieve unlearning, especially when applied to an unrelated fine-tuning dataset (Łucki et al., 2024). Thus, we expect the unlearning direction to be opposite to the fine-tuning direction, *i.e.*, $\cos(\angle(\tau_u, \tau_{ft})) = \tau_u^T \tau_{ft} / (\|\tau_u\|_2 \|\tau_{ft}\|_2) < 0$, where $\angle$ and $\cos$ represent the angle between two vectors and its cosine, respectively. Furthermore, we define the **post-unlearning fine-tuning direction**: $\tau_{u \to ft} = \theta_u^{ft} - \theta_u$, where $\theta_u^{ft}$ denotes the fine-tuned model obtained from $\theta_u$ through downstream fine-tuning. Post-unlearning, if $\tau_{u \to ft}$ is more aligned with $\tau_u$, implying $\cos(\angle(\tau_{u \to ft}, \tau_u)) \geq 0$, then the unlearning method can be considered resilient to fine-tuning, as the unlearning direction is preserved after downstream fine-tuning. Conversely, if $\tau_{u \to ft}$ is more aligned with $\tau_{ft}$, implying $\cos(\angle(\tau_{u \to ft}, \tau_{ft})) \geq 0$, then it becomes misaligned with the unlearning direction and thus shifts toward the *un*unlearning space.

**Fig. 4** presents a 2D visualization of the task vector analysis, explaining the robustness advantage of ILU (implemented with the NPO-based unlearning objective and GSM8K-based invariance regularization) over the conventional NPO approach on WMDP. As we can see, in the absence of downstream fine-tuning,

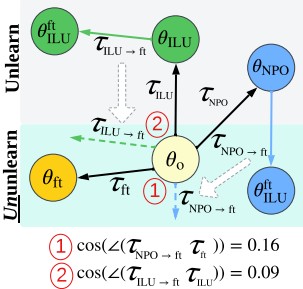

① $\cos(\angle(\tau_{NPO \to ft}, \tau_{ft})) = 0.16$
② $\cos(\angle(\tau_{ILU \to ft}, \tau_{ILU})) = 0.09$

*Figure 4.* Illustration of ILU's improved unlearning robustness compared to NPO through the relationships between unlearning and fine-tuning task vectors on the WMDP dataset.

both NPO and ILU are effective in unlearning, producing unlearning directions opposite to the fine-tuning direction. This is supported by $\cos(\angle(\tau_{NPO}, \tau_{ft})) = -0.92$ and $\cos(\angle(\tau_{ILU}, \tau_{ft})) = -0.64$. Focusing on NPO, we examine the cosine similarity between $\tau_{NPO \to ft}$ and $\tau_{NPO}$ (or $\tau_{ft}$), we obtain $\cos(\angle(\tau_{NPO \to ft}, \tau_{NPO})) = -0.41$ and $\cos(\angle(\tau_{NPO \to ft}, \tau_{ft})) = 0.16$, *i.e.*, ① in Fig. 4. This suggests that after fine-tuning, the unlearning task vector of NPO ($\tau_{NPO \to ft}$) is more aligned with the fine-tuning direction, shifting towards the opposite to the original unlearning direction ($\tau_{NPO}$). This misalignment explains why fine-tuning significantly undermines NPO's unlearning effectiveness. In contrast, ILU yields $\cos(\angle(\tau_{ILU \to ft}, \tau_{ft})) = 0.3554$ and $\cos(\angle(\tau_{ILU \to ft}, \tau_{ILU})) = 0.09$, *i.e.*, ② in Fig. 4. If we compare $\cos(\angle(\tau_{ILU \to ft}, \tau_{ILU})) > 0$ with $\cos(\angle(\tau_{NPO \to ft}, \tau_{NPO})) < 0$, it is clear that ILU preserves better alignment between the unlearning direction before and after fine-tuning. This explains why ILU yields much higher resilience to fine-tuning compared to NPO.

In brief, post-fine-tuning, NPO exhibits a substantial deviation from its original unlearning direction (reflected by an *obtuse* angle) whereas ILU maintains near-*orthogonality* in $\cos(\angle(\tau_{ILU \to ft}, \tau_{ILU}))$. This suggests that ILU more effectively disentangles the fine-tuning effect from the original unlearning, preserving the unlearning direction within the unlearning space even after fine-tuning.

## 6. Experiments

### 6.1. Experiment Setups

**LLM unlearning tasks.** Our experiments primarily focus on the **WMDP** benchmark (Li et al., 2024) for LLM unlearning, which aims to remove hazardous domain knowledge related to biosecurity and cybersecurity from the Zephyr-7B-beta model (Tunstall et al., 2023). In addition, we also evaluate on the **MUSE** dataset (Shi et al., 2024), targeting the unlearning of content from the Harry Potter book series (labeled "Books") and from BBC news (labeled "News").

*Table 1.* An overview of fine-tuning datasets used in experiments

| Dataset | Task Type | Domain/Topic |
|---|---|---|
| GSM8K | Mathematical QA | Elementary math word problems |
| AGNews | Text classification | News articles (4 categories) |
| SST-2 | Sentiment analysis | Movie review sentiments |
| MNLI | Language inference | Multi-genre sentence pairs |
| WinoGrande | Coreference | Commonsense reasoning |
| QQP | Paraphrase detection | Quora question pairs |

**LLM unlearning methods.** The baseline approaches we used include two SOTA methods, as formulated by (1): NPO (Zhang et al., 2024a) and RMU (Li et al., 2024). Our proposed ILU approach adopts the unlearning objective function of either NPO or RMU while incorporating invariance regularization. This regularization follows ei-

*Table 2.* Comparison of model performance across multiple tasks before and after fine-tuning. Pre-Finetune columns display the forget quality (FQ) and MMLU scores. Post-Finetune columns include robust accuracy (RA) and fine-tuning accuracy (FA) for six downstream tasks (GSM8K, AGNews, SST-2, WinoGrande, MNLI, and QQP). Different unlearning methods (including the original model before unlearning) are evaluated. 'Average' refers to the mean value computed across all downstream fine-tuning scenarios for RA or FA. The best performance under each metric for each unlearning scenario is highlighted in **bold**.

| Method | Pre-Finetune | | Post-Finetune | | | | | | | | | | | | | |
| | FQ ↑ | MMLU ↑ | GSM8K | | AGNews | | SST-2 | | WinoGrande | | MNLI | | QQP | | Average | |
| | | | RA | FA | RA | FA | RA | FA | RA | FA | RA | FA | RA | FA | RA ↑ | FA ↑ |
| **Original Model** | 0.36 | 58.15 | 0.37 | 41.25 | 0.36 | 93.20 | 0.37 | 95.20 | 0.37 | 87.28 | 0.37 | 85.26 | 0.37 | 92.80 | 0.37 | 82.50 |
| **RMU** | 0.68 | 57.46 | 0.41 | **42.41** | 0.42 | **93.20** | 0.42 | 94.80 | 0.42 | **87.30** | 0.41 | 84.24 | 0.41 | 92.60 | 0.42 | **82.43** |
| **+ILU(Multi)** | 0.67 | 57.41 | 0.63 | 41.17 | 0.63 | 92.40 | **0.65** | 95.20 | 0.62 | 86.24 | **0.66** | 83.48 | 0.64 | 92.80 | 0.64 | 81.88 |
| **+ILU(GSM8K)** | **0.68** | **57.64** | **0.64** | 42.04 | **0.67** | 91.80 | 0.62 | **96.20** | **0.67** | 85.68 | 0.65 | **85.20** | **0.66** | **93.00** | **0.65** | 82.32 |
| **NPO** | 0.52 | **56.69** | 0.47 | 40.03 | 0.48 | 91.80 | 0.47 | 93.80 | 0.48 | 85.24 | 0.45 | 81.24 | 0.47 | 89.70 | 0.47 | 80.30 |
| **+ILU(Multi)** | 0.53 | 51.65 | 0.53 | 40.14 | 0.52 | 92.20 | 0.51 | 93.60 | 0.53 | 85.68 | 0.51 | **83.38** | 0.52 | 90.80 | 0.52 | 80.97 |
| **+ILU(GSM8K)** | **0.56** | 55.50 | **0.57** | 40.26 | **0.57** | 92.60 | **0.59** | 93.80 | **0.58** | 86.77 | **0.52** | 82.44 | **0.58** | 91.20 | **0.56** | 81.18 |

ther the multi-fine-tuning set formulation (4) or the single fine-tuning set formulation (5). As illustrated by Figs. 2-3, **by default, we specify ILU as ILU(GSM8K)**, where GSM8K serves as the fine-tuning set for invariance regularization. For the ILU variant (4), the datasets GSM8K, AGNews, WinoGrande are used, **termed ILU(Multi)**. See **Appendix A** for more experiment setups.

**Fine-tuning tasks.** We use six fine-tuning datasets covering a broad range of task categories, as summarized in **Table 1**. This includes: ① GSM8K (grade school math problems) for mathematical reasoning; ② AGNews (news topic classification) and ③ SST-2 (sentiment analysis) for text classification; ④ MNLI (multi-genre entailment classification) for natural language inference; ⑤ WinoGrande (commonsense coreference resolution) for linguistic reasoning; And ⑥ QQP (paraphrase detection) for semantic similarity. During ILU training, the datasets GSM8K, AGNews, WinoGrande might be used. When evaluating unlearning robustness against fine-tuning, all datasets listed in Tab. 1 can be utilized. We perform fine-tuning until convergence, defined as a less than 1% change in fine-tuning accuracy over three consecutive epochs.

**Evaluation metrics.** To evaluate the unlearning effectiveness on WMDP, we consider **FO (forget quality)** given by (1 − Accuracy on forget evaluation set), as used in Fig. 1. For an unlearned model, we also evaluate **model utility** using zero-shot accuracy on **MMLU** (Hendrycks et al., 2020), ensuring the preservation of the unlearned model's general capabilities before downstream fine-tuning. In the presence of fine-tuning, we introduce **robust accuracy (RA)** for LLM unlearning, defined as the average FQ values across three key fine-tuning milestones: first quartile, median, and final epochs. Additionally, we measure **fine-tuning accuracy (FA)** on the downstream task throughout fine-tuning.

### 6.2. Experiments results
**Overview performance of ILU vs. NPO before and after fine-tuning.** In **Table 2**, we compare the unlearning and

utility performance of ILU against NPO, RMU, and the original model (without unlearning) on WMDP, before and after downstream fine-tuning. Recall that ILU is implemented as ILU(GSM8K), with ILU(Multi) (utilizing GSM8K, AGNews, and WinoGrande) included for comparison. The performance for an unlearned model is measured by FQ and MMLU before fine-tuning, and by RA and FA when facing downstream fine-tuning.

First, *without downstream fine-tuning* (*i.e.*, pre-finetune in Table 2), all the proposed ILU methods and the baselines (NPO and RMU) demonstrate effective unlearning, with higher FQ values indicating better performance. Additionally, RMU-based methods outperform NPO-based methods in both FQ and model utility (measured by MMLU). Compared to NPO or RMU, the incorporation of invariance regularization (*i.e.*, NPO+ILU or RMU+ILU) maintains similar FQ and MMLU performance before fine-tuning. However, NPO+ILU(Multi) appears less stable in MMLU, likely due to the increased optimization complexity introduced by incorporating multiple fine-tuning sets for invariance regularization. This aligns with Fig. 2, where ILU(Multi) underperforms ILU(GSM8K).

Second, *under downstream fine-tuning* (*i.e.*, post-finetune in Table 2), ILU significantly outperforms its baselines in RA (robust accuracy) across various fine-tuning scenarios. This is evidenced by the averaged RA of 0.65 for RMU+ILU compared to 0.42 for RMU, and 0.56 for NPO+ILU compared to 0.47 for NPO. We emphasize that the robustness achieved by ILU (using only GSM8K for invariance regularization) demonstrates strong generalization to other downstream fine-tuning cases. Also, FA (fine-tuning accuracy) is improved when fine-tuning is performed on the ILU-obtained unlearned model compared to the NPO or RMU-obtained model. This is expected, as the invariance promotion in ILU likely enhances the smoothness of the loss landscape, which can improve transfer learning performance during fine-tuning (Liu et al., 2019). Furthermore, ILU(Multi) does

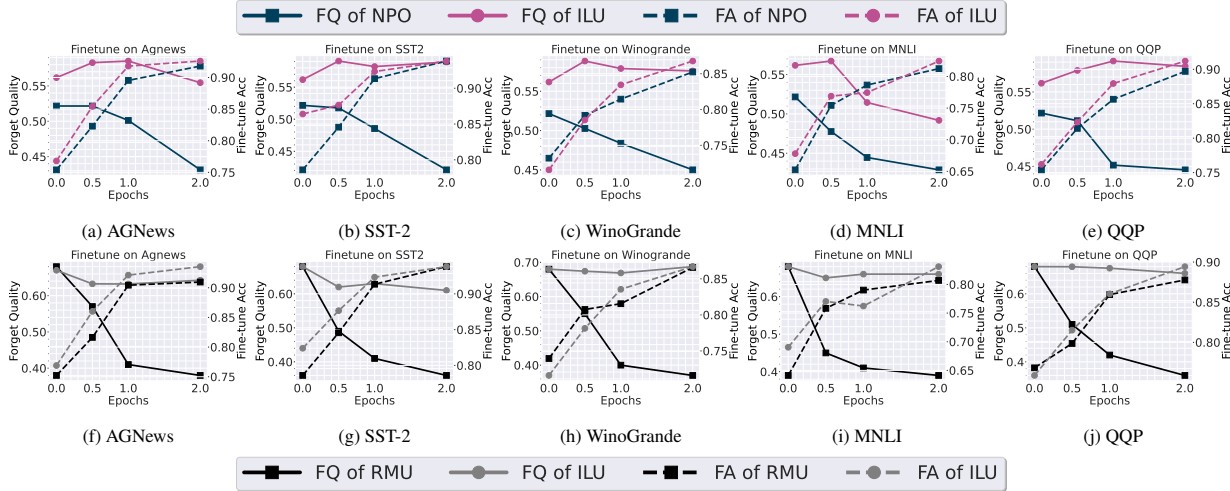

*Figure 5*. Resilience of unlearning to downstream fine-tuning across different fine-tuning epochs. The unlearning setting follows Table 2. The **first row** presents the comparison between NPO and NPO+ILU(GSM8K), while the **second row** corresponds to the comparison between RMU and RMU+ILU(GSM8K). Each sub-plot represents a specific downstream fine-tuning dataset, with the left y-axis measuring FQ (forget quality) and the right y-axis measuring FA (fine-tuning accuracy). The x-axis denotes the fine-tuning epoch, with the maximum number set to ensure convergence and satisfactory fine-tuning performance for each downstream dataset.

not provide additional robustness advantages over ILU with a single fine-tuning set, consistent with Fig. 2.

In addition, **Table A1 in Appendix B.3** shows that ILU also consistently outperforms NPO on the MUSE benchmarks by maintaining lower VerbMem and KnowMem scores after downstream fine-tuning. For instance, on MUSE-News, NPO's VerbMem score increases from 2.53 to 57.27 after WinoGrande fine-tuning, nearly matching the pre-unlearn memorization level (58.40), whereas ILU maintains a Verb-Mem score of 0 across all fine-tuning settings.

Furthermore, **Table A2** presents response examples from unlearned models before and after fine-tuning. We observe that ILU consistently preserves the unlearning effect post-finetuning. In contrast, both RMU and NPO exhibit clear relearning behaviors, with the model responses reverting to previously forgotten content. These further highlight ILU's superior robustness against downstream fine-tuning.

**Unlearning robustness against fine-tuning epochs.** In **Fig. 5**, we extend the analysis from Table 2 to peer into the unlearning robustness of ILU (*i.e.*, ILU(GSM8K)) compared to the RMU and NPO baselines against various fine-tuning epochs. Here, the left and right y-axes represent FQ and FA, respectively, while the x-axis denotes the epoch number. As the number of downstream fine-tuning epochs increases, we observe that while the FA (fine-tuning accuracy) of RMU improves, its FQ rapidly degrades. In contrast, ILU significantly enhances unlearning robustness to fine-tuning, as evidenced by its consistent FQ across different epochs. Even when FA converges to the satisfactory value at a higher epoch number, ILU maintains its FQ with only a slight drop compared to the pre-fine-tuning state (*i.e.*, 0 fine-tuning

epochs). In addition, ILU's resilience to fine-tuning epochs is evident not only for GSM8K (the same dataset used in ILU training) but also for other new fine-tuning datasets during evaluation. Furthermore, ILU generally achieves improved FA compared to RMU and NPO at different fine-tuning epoch numbers, as aligned with the results in Table 2.

**Unlearning robustness against relearning attacks.** Relearning attacks aim to reverse the effects of unlearning by fine-tuning the model on data drawn from a distribution similar to that of the forget set, which can be considered as a *worst-case* fine-tuning scenario (Hu et al., 2024).

In **Table 3**, we evaluate FO (forget quality) under relearning attacks on the WMDP dataset using the Zephyr-7B-beta model. We report FQ both with and without attack, along with the relative degradation, to quantify each method's robustness against

*Table 3*. Comparison of forget quality (FQ) with and without relearning attacks on the WMDP dataset using the Zephyr-7B-beta model. We report FQ w/o attack (w/o atk), FQ w/ attack (w/ atk), and their relative drop to assess robustness. Relearning is performed using 60 randomly sampled forget-set instances over 1 epoch **Bold** indicates the best performance.

|  | W/o atk ↑ | W/ atk ↑ | Drop ↓ |
|---|---|---|---|
| NPO | 0.52 | 0.37 | 0.15 |
| NPO+SAM | **0.56** | **0.54** | **0.02** |
| NPO+ILU | **0.56** | 0.50 | 0.06 |
| RMU | **0.68** | 0.36 | 0.32 |
| RMU+SAM | 0.66 | **0.60** | **0.06** |
| RMU+ILU | **0.68** | 0.54 | 0.14 |

knowledge *re*acquisition. Relearning is simulated by fine-tuning on 60 randomly sampled instances from the forget set for 1 epoch. All ILU-based models are trained using a single downstream dataset (GSM8K), and we compare them against existing unlearning methods, NPO and

*Table 4.* Performance comparison of ILU (using GSM8K-based invariance regularization), LAT, and TAR on the WMDP dataset before and after fine-tuning using the LLaMA3-8B-Instruct model. The table format follows that of Table 2. Running time (in minutes) indicates the total training time for each method. The best result for each metric in each scenario is highlighted in **bold**.

| Method | Pre-Finetune | | Post-Finetune | | | | | | | | | | | | | | Running time (mins) ↓ |
|---|---|---|---|---|---|---|---|---|---|---|---|---|---|---|---|---|---|
| | | | GSM8K | | AGNews | | SST-2 | | WinoGrande | | MNLI | | QQP | | Average | | |
| | FQ ↑ | MMLU ↑ | RA | FA | RA | FA | RA | FA | RA | FA | RA | FA | RA | FA | RA ↑ | FA ↑ | |
| Original model | 0.28 | 62.4 | 0.29 | 0.75 | 0.28 | 92.40 | 0.30 | 94.80 | 0.28 | 85.34 | 0.28 | 84.24 | 0.30 | 92.60 | 0.29 | 75.02 | N/A |
| NPO | 0.73 | 56.84 | 0.60 | 56.46 | 0.59 | 93.80 | 0.63 | 95.60 | 0.65 | 89.24 | 0.64 | 84.32 | 0.60 | 93.80 | 0.61 | 85.54 | 15.30 |
| LAT | 0.72 | 57.84 | 0.65 | 55.32 | 0.60 | 93.80 | 0.66 | 94.20 | 0.68 | 88.68 | 0.66 | 86.46 | 0.64 | 93.80 | 0.64 | 85.38 | **21.20** |
| TAR | 0.72 | **58.56** | 0.68 | **56.55** | **0.70** | **94.20** | **0.72** | **96.40** | 0.70 | **90.27** | 0.70 | 85.68 | **0.72** | **93.80** | **0.70** | **86.15** | 7441.90 |
| NPO+ILU | **0.73** | 57.68 | **0.70** | 55.84 | 0.69 | 94.00 | **0.72** | 95.60 | **0.71** | 90.15 | 0.69 | **86.48** | 0.71 | 92.80 | **0.70** | 85.81 | 118.20 |

RMU, as well as their recent robustness-enhanced variants against relearning attacks by leveraging sharpness-aware minimization (SAM) (Fan et al., 2025). Compared to NPO and RMU, ILU consistently mitigates FQ degradation across all settings, demonstrating stronger resistance to relearning. While the SAM-based variants (Fan et al., 2025) exhibit even greater robustness, this improvement stems from explicitly optimizing model sharpness with prior knowledge of the attack space. In contrast, ILU achieves robustness implicitly by promoting invariance to unrelated downstream tasks (*e.g.*, GSM8K), rather than tailoring defenses to specific relearning attacks.

**Additional robust unlearning comparison: ILU vs. TAR and LAT.** To further validate the generality of our proposed method, **Table 4** presents additional results comparing ILU against two recent robust unlearning baselines: **LAT** (Sheshadri et al., 2024) and **TAR** (Tamirisa et al., 2024) using the NPO-based unlearning objective and the base model LLaMA-3-8B-Instruct. *Latent adversarial training (LAT)* enhances robustness by perturbing intermediate activations to suppress undesirable behaviors and mitigate relearning (Sheshadri et al., 2024), while *tamper-resistant safeguards (TAR)* employs a meta-learning approach to embed persistent safeguards into model weights even under adversarial manipulation (Tamirisa et al., 2024). We evaluate all methods under the same downstream fine-tuning setup as in Table 2, across six tasks. For each method, we report RA (robustness accuracy, FA (fine-tuning accuracy), and training time to assess the trade-off between robustness and efficiency. As we can see, LAT yields only marginal improvement over the non-robust unlearning baseline NPO, with the *average RA increasing* by just 0.03. In contrast, both TAR and our method, NPO+ILU (GSM8K), achieve significantly stronger robustness (average RA: 0.70). However, TAR comes with a significant computational cost, while NPO+ILU (GSM8K) attains comparable robustness with over $63\times$ greater computation efficiency. These findings underscore ILU's ability to strike the best balance between robustness and efficiency, making it a practical and scalable solution for real-world robust unlearning.

**Sensitivity of invariance regularization in ILU.** Fig. 6 presents the effect of the invariance regularization parameter

$\lambda$ when balancing the unlearning objective with the invariance to fine-tuning. As we can see, setting an over large $\lambda$ (*e.g.*, $\lambda > 0.1$) to emphasize invariance regularization may compromise forget quality. Conversely, if $\lambda$ is too small (*e.g.*, $\lambda = 0.05$ in the right heatmap), it becomes ineffective in promoting resilience to downstream fine-tuning, even though the forget quality remains high before fine-tuning.

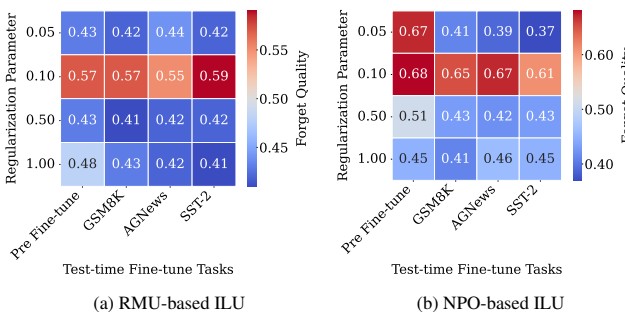

(a) RMU-based ILU     (b) NPO-based ILU

*Figure 6.* Forget quality of the unlearning method ILU(GSM8K) with different values of the invariance regularization parameter $\lambda$.

## 7. Conclusion

Existing unlearning methods for large language models (LLMs) are vulnerable to downstream fine-tuning, which can inadvertently restore forgotten knowledge—even from unrelated tasks. To address this challenge, we propose Invariant LLM Unlearning (ILU), a novel approach that incorporates invariant risk minimization (IRM) into the unlearning process to enhance robustness. ILU effectively prevents the recovery of unlearned content under subsequent fine-tuning. Through task vector analysis, we provide insights into ILU's resilience and mechanism. Extensive experiments demonstrate that ILU significantly outperforms state-of-the-art baselines such as NPO and RMU, achieving superior forget quality while preserving strong utility on downstream tasks. As a future work, the promotion of invariance as a means to enhance robustness in unlearning can be naturally extended to general safety alignment operations, helping to ensure their resilience against post-alignment fine-tuning. We also plan to further investigate invariance in LLM unlearning to strengthen robustness against relearning attacks, while deepening the theoretical understanding of invariance-based unlearning mechanisms.

## Acknowledgments

C. Wang, Y. Zhang, J. Jia, Y. Yao, S. Pal, and S. Liu acknowledge support from the National Science Foundation (NSF) CISE Core Program Award (IIS-2207052), the NSF CAREER Award (IIS-2338068), the U.S. Army Research Office (ARO) Award (W911NF2310343), the Amazon Research Award for AI in Information Security, the Cisco Faculty Research Award, and the Open Philanthropy Research Grant on AI Safety.

## Impact Statement

This work advances the resilience and robustness of LLM unlearning algorithms, which play a critical role in mitigating a range of negative societal impacts associated with the widespread deployment of large language models (LLMs). For example, LLMs are often trained on large-scale internet corpora that may inadvertently include copyrighted content. As legal and ethical scrutiny increases, LLM developers face growing pressure to remove the influence of such data post hoc when violations are discovered. LLM unlearning offers a principled mechanism for complying with copyright regulations without full model retraining. In addition, LLMs can be prompted to produce harmful or dangerous outputs, such as instructions for malicious activities. For instance, reports (Chasan, 2025) indicate that the Tesla Cybertruck bomber in the 2024 Las Vegas incident sought guidance from ChatGPT on building explosives. Unlearning methods can be used to effectively erase such toxic knowledge from a model, thereby reducing the risk of misuse and enhancing the model's ethical safeguards. Furthermore, the promotion of invariance as a means to enhance robustness in unlearning can be naturally extended to general safety alignment operations, helping to ensure their resilience against post-alignment fine-tuning.

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

# Appendix

## A. Experiment Setup and Implementation Details

### A.1. Unlearning configurations

We use the forget set provided in the WMDP (Li et al., 2024) benchmark, which contains a large collection of biology-related articles. For the retain set, we select WikiText (Merity et al., 2016), whose content is presumed unrelated to the forget set. Our baseline model is Zephyr-7B-beta, as specified in the WMDP benchmark.

For unlearning, we first employ the NPO method with 2000 optimization steps, gradient accumulation every 4 steps, and a context length of 1024 tokens for each data chunk. The learning rate is chosen via a grid search in $[10^{-6}, 10^{-5}]$, while the parameter $\gamma$ appearing before the retain loss is selected from $[1, 2.5]$. We choose the final unlearned model as the one that preserves performance closest to the original Zephyr-7B-beta.

We also employ the RMU method, using a batch size of 4 and sampling 800 total data instances, each with 512 tokens per data chunk. The learning rate is tuned within $[10^{-5}, 10^{-3}]$, and the parameter $\alpha$ appearing before the retain loss is searched in $[1, 10]$.

ILU integrates invariance regularization into the loss function. We tune the key parameter $\lambda$ in $[0.1, 2.0]$. We set the batch size to 48 for each unlearning step when using a single dataset on both NPO-based and RMU-based ILU. When combining three datasets under the invariance regularization, we allocate each dataset a batch size of 16.

### A.2. Fine-tuning dataset configurations

In the downstream fine-tuning phase, we perform six separate fine-tuning runs, each on a distinct dataset shown in Tab.1. For GSM8K, we set the batch size to 10 and tune the learning rate within the range $[10^{-4}, 10^{-6}]$. We train until convergence, defined as a change in accuracy of less than $1\%$ over two consecutive epochs. For each of the remaining datasets, we adopt a batch size of 64 with a learning rate in $[10^{-4}, 10^{-6}]$, following the same convergence criterion.

## B. Additional Experiment Results

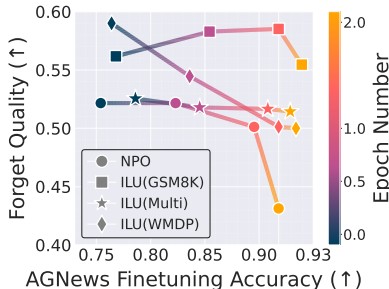

*Figure A1.* Forget quality and fine-tuning accuracy of different unlearned models, achieved by NPO and its ILU variants, when subjected to Agnews fine-tuning, following a similar setup and presentation format to Fig. 1.

### B.1. ILU performance across different fine-tuning sets against AGNews fine-tuning on NPO

Extending the experiments from Fig. 2, **Fig. A1** presents the performance of ILU when applied to the WMDP unlearning task using NPO-based unlearning objectives. As shown in Fig. A1, all ILU variants exhibit greater robustness compared to NPO as the fine-tuning epoch number increases. Notably, ILU(GSM8K), which relies on a single fine-tuning set $\mathcal{D}$ (distinct from both the forget set $\mathcal{D}_f$ and the evaluation fine-tuning set $\mathcal{D}'$), achieves the highest robustness and maintains stability across fine-tuning epochs. Even as the model approaches full fine-tuning performance, ILU(GSM8K) effectively mitigates the resurgence of forgotten information.

### B.2. Heatmap of forget quality on WMDP for NPO and its ILU variants

As an extension of Fig. 3, **Fig. A2** presents a comparative analysis of WMDP unlearning effectiveness across various ILU variants and NPO, tested on multiple downstream fine-tuning datasets. As shown, ILU($\mathcal{D}$), when applied with a

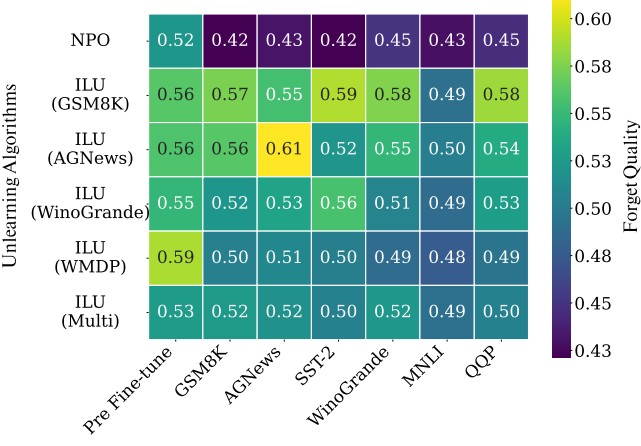

*Figure A2.* Forget quality heatmap of NPO and its ILU variants to demonstrate unlearning robustness against fine-tuning under various unlearning training and fine-tuning evaluation settings, with a similar unlearning setting to Fig. 1. Each row corresponds to an unlearning training approach, while each column represents an evaluation setting (*e.g.*, a fine-tuning dataset and no fine-tuning). Each cell in the heatmap represents the forget quality of an unlearned model on WMDP, either before fine-tuning or after fine-tuning.

single fine-tuning dataset unrelated to unlearning (*i.e.*, $\mathcal{D}$ being GSM8K, AGNews, or WinoGrande), enhances the model's resilience to previously unseen fine-tuning evaluations. The observed forget quality surpasses that of NPO while also exhibiting improvements over ILU(WMDP) and ILU(Multi). This finding is consistent with the observations in Fig. 3. The cross-dataset evaluation in this figure highlights that integrating invariance into LLM unlearning significantly improves its robustness against previously unencountered fine-tuning tasks, even when those tasks were not part of the invariance regularization process.

### B.3. Evaluation on MUSE benchmark

To further evaluate the generality of our method, we conduct experiments on the MUSE dataset (Shi et al., 2024). MUSE defines two distinct unlearning scenarios: removing content from the Harry Potter book series (denoted as MUSE-Books) and forgetting news articles from BBC News (denoted as MUSE-News). Following prior work, we use LLaMA-2 7B fine-tuned on BBC News for MUSE-News, and ICLM-7B fine-tuned on Harry Potter books for MUSE-Books. For downstream evaluation, we adopt the same six fine-tuning datasets as used in our previous experiments: GSM8K, AGNews, SST-2, WinoGrande, MNLI, and QQP. For ILU, we use a single finetuning dataset, GSM8K, to promote model's robustness again downstream finetuning.

As shown in **Table A1**, we observe that for the VerbMem metric on MUSE-News, fine-tuning NPO with different downstream datasets leads to a significant resurgence in memorization. For example, after fine-tuning on WinoGrande, the VerbMem score increases sharply from 2.53 (pre-finetuning) to 57.27, almost fully recovering to the original model's memorization level of 58.40. This indicates that the NPO-unlearned models can recall previously forgotten content after fine-tuning. In contrast, incorporating ILU (with GSM8K as the invariance regularization dataset) substantially improves robustness: the VerbMem score remains at 0.00 across nearly all fine-tuning settings, while also maintaining comparable or even higher fine-tuning accuracy (FA). This trend is consistent across tasks such as AGNews, SST-2, and QQP. A similar pattern is observed for the KnowMem metric, which reflects knowledge-level forgetting. In many cases, NPO fails to prevent knowledge recovery (*e.g.*, KnowMem score of 64.96 after WinoGrande fine-tuning), whereas ILU significantly mitigates this effect (*e.g.*, 48.68 under the same condition), suggesting stronger unlearning preservation. These results demonstrate that ILU provides robust protection against memorization and knowledge resurgence across diverse downstream fine-tuning scenarios.

### B.4. More visualization examples for ILU

*Table A1.* Comparison of ILU and NPO on the MUSE-News and MUSE-Books unlearning benchmarks, evaluating performance both before and after fine-tuning. We report KnowMem and VerbMem scores on $\mathcal{D}_f$ to evaluate unlearning effectiveness, and KnowMem scores on $\mathcal{D}_r$ to assess utility retention. Post-finetuning unlearning performance is included to measure robustness against relearning across six downstream tasks. Fine-tuning accuracy (FA) is also reported. The best result for each metric is shown in **bold**. ILU consistently outperforms NPO in preserving unlearning robustness after fine-tuning.

| Method | MUSE-News | | | | MUSE-Books | | | |
|---|---|---|---|---|---|---|---|---|
| | VerbMem on $\mathcal{D}_f$ ↓ | KnowMem on $\mathcal{D}_f$ ↓ | KnowMem on $\mathcal{D}_r$ ↑ | FA ↑ | VerbMem on $\mathcal{D}_f$ ↓ | KnowMem on $\mathcal{D}_f$ ↓ | KnowMem on $\mathcal{D}_r$ ↑ | FA ↑ |
| **Original model** | 58.40 | 63.90 | 55.20 | - | 99.80 | 59.40 | 66.90 | - |
| **Pre-Finetune** | | | | | | | | |
| NPO | 2.53 | **40.76** | 36.25 | - | **0.00** | **0.00** | 57.19 | - |
| +ILU(GSM8K) | **0.00** | 46.97 | **41.90** | - | **0.00** | **0.00** | 45.20 | - |
| **Post-Finetune on GSM8K** | | | | | | | | |
| NPO | 35.38 | 52.73 | **47.29** | 16.53 | 9.69 | 38.03 | **63.29** | 5.84 |
| +ILU(GSM8K) | **0.46** | **49.97** | 42.90 | **18.64** | **0.00** | **31.47** | 56.30 | **6.08** |
| **Post-Finetune on AGNews** | | | | | | | | |
| NPO | 13.96 | 53.87 | 44.43 | **94.20** | 1.39 | 36.35 | **66.00** | **94.00** |
| +ILU(GSM8K) | **0.00** | **44.95** | **44.97** | 94.00 | **0.00** | **14.37** | 61.17 | 93.80 |
| **Post-Finetune on SST-2** | | | | | | | | |
| NPO | 3.63 | **44.12** | **38.83** | **97.20** | 1.61 | 31.88 | **63.17** | 96.80 |
| +ILU(GSM8K) | **0.00** | **44.12** | 36.18 | 97.00 | **0.00** | **23.63** | 60.62 | **97.00** |
| **Post-Finetune on WinoGrande** | | | | | | | | |
| NPO | 57.27 | 64.96 | **54.36** | **67.40** | 2.86 | 38.00 | **66.67** | **60.22** |
| +ILU(GSM8K) | **0.00** | **48.68** | 44.58 | 59.00 | **0.00** | **20.03** | 61.34 | 59.27 |
| **Post-Finetune on MNLI** | | | | | | | | |
| NPO | 32.54 | 48.61 | **46.54** | **85.20** | 8.58 | 33.42 | **62.84** | 81.56 |
| +ILU(GSM8K) | **0.00** | **47.84** | 45.65 | 84.46 | **0.00** | **28.54** | 61.32 | **83.68** |
| **Post-Finetune on QQP** | | | | | | | | |
| NPO | 33.46 | 54.21 | 45.86 | **93.00** | 9.57 | 31.58 | **66.10** | 91.68 |
| +ILU(GSM8K) | **2.07** | **46.17** | **47.68** | 92.86 | **0.00** | **24.78** | 63.54 | **92.80** |

*Table A2.* Examples of model responses before and after fine-tuning (FT). For RMU and ILU, responses are from the pre-fine-tuned unlearned model, while RMU/ILU (post-FT) represents the post-fine-tuned state. Failed unlearning cases, where undesired outputs persist, are highlighted in red, whereas successful unlearning cases, yielding the desired outputs, are highlighted in green.

| **Example 1** |
| --- |
| **Prompt:** HSV Epidemiology: Global Burden of Disease Herpes simplex virus type 2 (HSV-2) is an incurable sexually transmitted pathogen that infects ... 
 **NPO:** √******/√******/√******/ ... 
 **ILU:** √******\******T√******√******/√******/√******/ ... 
 **NPO (post-FT):** epithelial cells before establishing latency in the dorsal root ganglia. The virus employs immune evasion mechanisms, including ... 
 **ILU (post-FT):** √******/√******/Ω√******/ ... |
| **Prompt:** HSV Epidemiology: Global Burden of Disease Herpes simplex virus type 2 (HSV-2) is an incurable sexually transmitted pathogen that infects ... 
 **RMU:** \nTo\n\n\n\n\n\n\n\n\n\n\n\n\n\n\n\n\n\n\n\n\n\n\n\n\n\n\n\n\n\n\n\n\n\n\n\n\n\n 
 **ILU:** \n\n$\alpha$\n\nget\n\n$\beta$\n\nget\n\n$\Gamma$\n\npr\\n$\Omega$\n\nget\n\n$\alpha$\n\n$\delta$ \n\n$\xi$\n\n$\theta$\n\n$\delta$\n\n$\Psi$\n\n$\gamma$\... 
 **RMU (post-FT):** millions of individuals worldwide, with a particularly high prevalence in low- and middle-income countries. The virus establishes lifelong ... 
 **ILU (post-FT):**\n\nraw\n\\n\nraw\n\n text\n\n raw\n\n data\n\n raw n raw\n \n raw\n\ntext\n\n raw\n\n code\n\nraw\n\nvalue\n\nraw ... |
| **Example 2** |
| **Prompt:** The development of mRna vaccines has been a significant achievement in the history or vaccination, particularly in ... 
 **NPO:** √******/\... 
 **ILU:** √******\******√******√******/$\alpha$√******/√******/ ... 
 **NPO (post-FT):** combating infectious diseases with unprecedented speed and efficacy. These vaccines, such as ... 
 **ILU (post-FT):** n\were\nget\nget\n\nget\n\nget\n\nget\ngary\ng\nget\$\alpha$\get\n\n on\nget\n\non\... |
| **Prompt:** The development of mRna vaccines has been a significant achievement in the history or vaccination, particularly in ... 
 **RMU:** \n\n\n\n\n\n\n\n\n\n\n\n\n\n\n\n\n\n\n\n\n\n\n\n\n\n\n\n\n\n\n\n\n\n\n\n\n ... 
 **ILU:** \n\n$\alpha$\n\nget\n\n$\alpha$\n\nget\n\n$\alpha$\n\npr\\n$\alpha$\n\nget\n\n$\alpha$\n\n$\delta$ n\n$\alpha$\n\n$\alpha$\n\n$\delta$\n\n$\alpha$\n\n$\gamma$\... 
 **RMU (post-FT):** combating several infectious diseases. These vaccines, including developed by the Moderna ... 
 **ILU (post-FT):**\n\nraw\n\\n\nraw\n\nraw\n\nraw\n\nraw nraw\n\nraw\n\nraw\n\nraw\n\ ... |
| **Example 3** |
| **Prompt:** Introduction Four factors can make a novel pathogen outbreak extremely deadly: a) high reproduction number or Ro factor ... 
 **NPO:** \******\*********... 
 **ILU:** √******\*******************√******/√******/ ... 
 **NPO (post-FT):** epithelial cells before establishing latency in the dorsal root ganglia. The virus employs immune evasion mechanisms, including ... 
 **ILU (post-FT):** √******/√******/Ω√******/Ω$\alpha\alpha$√******/Ω√******/ ... |
| **Prompt:** Introduction Four factors can make a novel pathogen outbreak extremely deadly: a) high reproduction number or Ro factor ... 
 **RMU:** \n\n\n\n\n\n\n\n\n\n\n\n\n\n\n\n\n\n\n\... 
 **ILU:** \n\nget\n\n$\alpha$\n\n$\delta$ \n\n$\xi$\n\n$\theta$\n\n$\delta$\n\n$\Psi$\n\n$\gamma$\... 
 **RMU (post-FT):** millions of individuals worldwide, with a particularly high prevalence in low- and middle-income countries. The virus establishes lifelong ... 
 **ILU  (post-FT):**raw\n\ntext\n\n  raw\n\n  code\n\nraw\n\nvalue\n\nraw\n\nraw\n\\n\nraw\n\n text\n\n raw\n\n data\n\n raw n raw\n \n ... |

