# OpenReview forum: "Invariance Makes LLM Unlearning Resilient Even to Unanticipated Downstream Fine-Tuning"
_ICML.cc/2025/Conference — ICML 2025 poster_

### Official Review · Reviewer_BD7R · 2025-03-04

**Overall Recommendation:** 3

**Summary:**

Incorporating an invariance objective into unlearning tasks can prevent accidentally relearning unlearned WMDP when trained on standard fine-tuning datasets.

**Claims And Evidence:**

While there is some evidence that the method is robust against relearning using unrelated fine-tuning tasks, I don’t think there is enough evidence across other models and unlearning settings and relearning attacks using WMDP itself to be able to say for sure that this is a significant contribution especially when other popular unlearning baselines designed to solves this problem are neglected.

**Essential References Not Discussed:**

I have mentioned these references earlier.

**Experimental Designs Or Analyses:**

There are several flaws in this work regarding experimental design related to a limited evaluation. First, even though TAR [1] is mentioned several times in the paper as a solution that was designed to provide robust fine-tuning it is neglected as a baseline. [2] also is a method that tries to treat this issue. [3] has provided a method for this as well though not in unlearning so I don’t expect the authors to add (though [2] did use it). For fair evaluation I ask that the authors select baselines that were designed to address the issue of robust unlearning.

[1] Tamirisa, R., Bharathi, B., Phan, L., Zhou, A., Gatti, A., Suresh, T., ... & Mazeika, M. (2024). Tamper-resistant safeguards for open-weight llms.

[2] Gandikota, R., Feucht, S., Marks, S., & Bau, D. (2024). Erasing conceptual knowledge from language models.

[3] Rosati, D., Wehner, J., Williams, K., Bartoszcze, L., & Gonzales, R. carsten maple, Subhabrata Majumdar, Hassan Sajjad, and Frank Rudzicz.(2024).  Representation noising: A defence mechanism against harmful finetuning

Second, even though the authors claim that WMDP is more realistic setting than TOFU, MUSE, or the harry potter dataset - It is still important to provide at least some other datasets to demonstrate the reliability of the newly proposed method. So I ask that at least one other unlearning dataset is added.

Third, although the relearning attack is discussed and in my reading of “Robustness challenge in LLM unlearning” is made a focal point of unlearning robustness, the results of this attack are not presented. Please present the results of this attack as it seems critical to being able to evaluate the utility of this method.

Fourth, I am concerned that only one model is used to demonstrate the efficacy of the method.I

**Methods And Evaluation Criteria:**

The authors add a gradient norm penalty to enhance previously existing unlearning methods. It makes sense that this works because the unlearned model is now much more stable in the loss landscape of the dataset used for the grad norm penalty. This is a valuable way to enhance previous methods and in the limited setting the authors do evaluate show that it is a promising approach.

The robust accuracy for unlearning is an interesting evaluation criteria. The evaluations that were performed make sense and are sound.

**Other Comments Or Suggestions:**

“ununlearning” is very confusing. I think in the literature relearning is preferred.

A.2. Which optimizer is used and what scheduler? I think this can matter a lot for robust unlearning (see [1])

[1] Deng, J., Pang, S., Chen, Y., Xia, L., Bai, Y., Weng, H., & Xu, W. (2024, May). Sophon: Non-fine-tunable learning to restrain task transferability for pre-trained models.

**Other Strengths And Weaknesses:**

To be clear, I find the actual method itself to be well validated within the scope of the experiments done. I also think this is valuable contribution and extensions. I just can’t recommend for acceptance until the reliability of the method is shown.

**Questions For Authors:**

None

**Relation To Broader Scientific Literature:**

While unlearning had its moment last year when it was adapted to the LLM setting from computer vision, there is currently disillusionment about the utility of unlearning while it remains vulnerable to easily performed attacks and even by accident. This work provides a valuable unique method that supplements other on going works attempting to make unlearning robust.

**Theoretical Claims:**

No derivation is provided for (3), it is unclear why (4) and then (5) serves as a reasonable approximation that satisfies (2). Please provide the full derivation and proof of optimality of the approximations. I’m not sure what non-stationarity and (269) stationarity are meant to refer to.

Section 5 is speculation without theoretical evidence.  It isn’t clear to me without proof that “fine-tuning direction resides in the ununlearning space” and  “we expect the unlearning direction to be opposite to the fine-tuning direction.” While Illharco et al 2023 is a nice empirical work, *linear* task directions are an unproven conjecture so the analysis doesn’t have much solid ground to go off. I recognize the empirical illustration with actual measurements but I am not convinced without a more robust experiment (variety of ft and unlearning datasets as well as more algorithms) to validate the conjecture and without theoretical justification. Even were this the case, I would hesitate to say its what explains ILU is more resilient compared to NPO. For example, an alternative explanation could be that the gradient regularization performed in (5) results in converging to a gradient 0 point i.e. trapped at a fixed point which means that gradient descent isn’t likely to succeed.

Finally, I find the motivation from IRM to be confusing and not well motivated. It isn’t clear to me what use this framework has when we (A) we don’t actually end up needed (multi) invariance to many datasets, (B) the advantage of IRM is when we have w be a model that uses the invariant theta but this isn’t done as w=1 (C) invariance during prediction (what IRM is used for) and invariance during training are two very different things: I think the claim that we can link invariance to robustness against fine-tuning isn’t shown.  It isn’t clear to me why IRM should be used over a more simpler motivation from gradient norm.

---

> ### Author Rebuttal · Authors · 2025-04-01
>
> **Q1: Derivation on (3), connection with (4)-(5), question on stationarity, motivation on IRM, and why not just gradient norm.**
>
> **A1:** Eq. (3) follows the standard IRMv1 relaxation from [R1], which approximates the bi-level IRM in Eq. (2) using a single-level gradient penalty. This promotes invariance across environments by encouraging stationarity (i.e., local optimality) of the shared predictor $\mathbf{w}$ across datasets $\mathcal{D}_i$. For a complete derivation, see Section 3.1 of [R1]. Stationarity means that $\nabla_w \ell_i(w \circ \theta) \approx 0$ for all $i$, implying $w=1$ is optimal across environments. The $w=1$ can be fixed as a scalar without loss of generality, as shown in [R1]. Our ILU objective (Eqs. 4–5) extends Eq. (3) to LLM unlearning by replacing empirical risk with unlearning loss and treating $\mathcal{D}_i$ as fine-tuning environments. The gradient norm penalty thus ensures that $\boldsymbol{\theta}$ remains locally optimal across tasks, making it robust to further fine-tuning. Hence, using the gradient norm is not arbitrary—it directly follows from IRM and supports the desired invariance.
> > [R1] arXiv:1907.02893.
>
> **Q2: Concerns about the lack of theoretical support in Section 5 and the speculative nature of linear task direction analysis.**
> **A2:** First, we admit that our task vector analysis is not a theoretical proof, but we believe it is grounded in practical relevance. In Lines 277–280, the task vector represents the direction from the fine-tuned model to the pre-trained model. We do not claim this as the precise gradient direction (i.e., no linear assumption is imposed). The use of task vectors to investigate task-wise learning direction geometry has been applied in non-linear cases [R2-4]. Even under a linear assumption, the work in [R3] demonstrates that this is a valid approach for practical task vectors ([Eq. (3), R3]). Additionally, recent work [R4] provides a theoretical proof for the general task vector in model editing tasks.
> > [R2]  arXiv:2212.04089
> > [R3]  arXiv:2305.12827
> > [R4]  https://openreview.net/forum?id=vRvVVb0NAz
>
> **Q3: Gradient regularization performed in (5) results in converging to a gradient 0.**
> **A3:** As addressed in Q1, the gradient norm is defined based on the fine-tuning datasets and aims to promote the universal optimality of unlearned model on these datasets, originating from IRM. As a result, ILU does not hinder fine-tuning success (i.e., it avoids getting trapped in a poor local optimum), as demonstrated in Table 2 and Fig. 5. This is further confirmed by the task vector analysis, where ILU shows better alignment between pre- and post-fine-tuning directions, while NPO causes conflicts, as showed in Q2.
> **Q4: We don’t actually end up needing (multi) invariance to many datasets.**
> **A4:** For details on why ILU stays effective and does not require (multi) invariance to many datasets, please refer to Q4 from [Reviewer yxbz](https://openreview.net/forum?id=x2lm33kdrZ&noteId=NjDB2xVUnJ).
> **Q5: Invariance during prediction and invariance during training are two things.**
> **A5:** Thank you for the comment. We believe there exists a misunderstanding of IRM.  We agree that prediction-time invariance and training-time robustness are conceptually different. However, in IRM, **training-time invariance** is the mechanism used to achieve prediction-time invariance. Specifically, IRM aims to learn a representation $\phi$ such that a single predictor $w$ performs optimally across environments (as formulated by lower-level training problem over $w$ in Eq. 2). This is enforced during training by penalizing the gradient of the environment-specific loss with respect to $w$, ensuring $w$ remains stable across environments. Our work extends this training-time mechanism to promote parameter-level invariance in the unlearned model $\theta$.
>
> **Q6: Several flaws in this work regarding experimental design.**
> **A6:** We are happy to add more experiments to strengthen our work; However, we respectfully disagree with the criticism that our current experimental design has "flaws." First, we have added an experiment comparing ILU with TAR (**[Table R3](https://ibb.co/4ZSkRQGh)**). Second, we discuss the suggested works [2] ("Erasing Conceptual Knowledge from Language Models") and [3] ("Representation Noising: A Defense Mechanism Against Harmful Fine-tuning"), however, we note that these works are quite different from ours. Specifically, [2] investigates robustness against input-level jailbreaking attacks, whereas our focus is on model-based fine-tuning. Third, we evaluated robustness against the relearning attack (**[Table R2](https://ibb.co/DHd5Zw2c)**),  Fourth, we extended experiments to the MUSE dataset (**[Table R4](https://ibb.co/m5Rtqc3t)**) Lastly, we confirmed ILU's effectiveness across different LLMs and datasets; See our detailed responses from [Reviewer b9VT](https://openreview.net/forum?id=x2lm33kdrZ&noteId=aitszercwQ)’s Q1–Q3.

---

> > ### Comment · Reviewer_BD7R · 2025-04-02
> >
> > Re:A1 - Thanks for this, I was missing the connection to stationarity across losses, optimality makes sense now!
> >
> > I really appreciate the effort put into A6. Largely my concerns over external validity and reliability of the study are satisfied. I am changing my score to a weak accept. As with other reviewers, I am not sure I am comfortable with a higher score without seeing more effort for expanding reliability and external validity of the study, there exist more unlearning methods e.g. listed in [1] and datasets that are still not evaluated. To make this more concrete seeing one or two more methods and evaluation on TOFU would satisfy me and I ask the authors add these in the camera ready.
> >
> > [1] Che, Z., Casper, S., Kirk, R., Satheesh, A., Slocum, S., McKinney, L. E., ... & Hadfield-Menell, D. (2025). Model Tampering Attacks Enable More Rigorous Evaluations of LLM Capabilities.
> >
> > I will repeat for the sake of the Area Chair that I do think that this is a valuable contribution and despite my concerns about the method still not rigourously evaluated the work is quite good.

---

> > > ### Author Response · Authors · 2025-04-05
> > >
> > > Thank you very much for recognizing our rebuttal efforts and raising the score.
> > >
> > > We also appreciate you pointing out [1] (a concurrent work, available on Feb. 2025) and suggesting we check other robust unlearning methods used in [1]. Per your suggestion, we carefully checked [1], which focuses on unlearning robustness evaluation on WMDP. The first thing we want to note is that TAR appears to be the strongest unlearning method in [1]. We already compared TAR with our approach in our earlier response, showing similar robustness gains but with significantly improved computational efficiency (see our previous response to [Q6](https://openreview.net/forum?id=x2lm33kdrZ&noteId=iyqcrAlefc) and response to [b9VT’s Q2](https://openreview.net/forum?id=x2lm33kdrZ&noteId=jtqu4A1bls)).
> > >
> > > Additionally, thank you for encouraging us to add TOFU experiments in the camera-ready version; We will make our best effort to do so. We value this suggestion, but we would like to respectfully note that the MUSE dataset used in our rebuttal shares a similar spirit to TOFU. Both aim to remove "factual" information from the forget set (fictional author information in TOFU, and copyrighted information in MUSE). We preferred MUSE to TOFU in the rebuttal because TOFU uses a p-value-based forget quality metric to evaluate unlearning effectiveness, which may not be very precise unless the p-value is close to 1 (good unlearning) or less than 0.05 (poor unlearning). A p-value greater than 0.05 (but away from 1) is not a precise metric for indicating “good” unlearning quality.
> > >
> > > Thank you once again for recognizing the valuable contribution of our paper and recommending it to the AC. We truly appreciate your thoughtful feedback and the time you’ve dedicated to reviewing our work！

---

### Official Review · Reviewer_yxbz · 2025-03-11

**Overall Recommendation:** 2

**Summary:**

The paper introduces a novel approach to enhance the resilience of LLMs against the re-emergence of unlearned knowledge during downstream fine-tuning tasks. This is achieved through invariant LLM unlearning, which incorporates IRM principles into the unlearning process. The contributions of this paper are summarised as follows:

LLM Unlearning Challenges: Current unlearning methods are effective but typically fail when the model undergoes downstream fine-tuning. This can inadvertently recover the knowledge that was supposed to be unlearned, posing a significant challenge in scenarios where LLMs are fine-tuned for new tasks post-unlearning.

Invariant Risk Minimization: The paper leverages IRM, a technique used for training models to perform well across varied environments, to ensure that the unlearning remains robust even after subsequent model fine-tuning. IRM helps enforce that the model's predictions remain invariant, regardless of changes in data distribution or task.

Invariant LLM Unlearning: ILU is proposed as a new framework that integrates IRM into the unlearning process. This approach aims to make the erased knowledge robustly inaccessible, even if the model is later adapted or fine-tuned for other tasks.

Empirical Validation: The effectiveness of ILU is demonstrated through extensive experiments on the WMDP benchmark, which focuses on removing harmful knowledge from LLMs. ILU significantly outperforms existing state-of-the-art unlearning methods like NPO and RMU, particularly in maintaining unlearning robustness across various fine-tuning scenarios.

Task Vector Analysis: The paper also introduces a task vector analysis to further understand how ILU maintains robustness. This analysis shows that ILU helps align the unlearning direction in a way that resists shifts caused by downstream fine-tuning, essentially preventing the relearning of unwanted knowledge.

This work not only advances the field of machine learning by addressing a significant shortcoming in LLM unlearning but also sets a foundational approach for future research into creating more resilient machine learning models against dynamic operational environments.

**Claims And Evidence:**

I think the authors do not fully explain why the scenario of relearning occurs, it can also be the reason why the parts to motivate ILU at the beginning in Section 4 seems weak to me.

I agree that NPO and RMU have the risks of relearning, but it may be mainly due to their under-unlearning. I am not sure if using GA can also lead to the scenario of relearning. Also, I agree that GA will lead to over-unlearning and the deterioration of retention, but I think you can consider some of its improved version, such as task vector, and wga [1].

[1] Rethinking LLM Unlearning Objectives: A Gradient Perspective and Go Beyond.

I am not sure the operation ⊥ is well defined and if we can formally claim that GSM8K is completely orthogonal to WMDP.

From my understanding, using the gradient direction to understand the influence on model performance works under linear assumption, which seems to correct only if the magnitude of \tau is quite small.  So, though quite interesting, I am not sure if the analysis in Section 5 is solid enough.

**Essential References Not Discussed:**

Good Enough.

**Experimental Designs Or Analyses:**

It seems that the empirical results are limited to WMDP. More results on TOFU and MUSE can obviously make the results more solid, but I think it is proper for now in conducting experiments only on WMDP. Moreover, the baseline methods are limited to RMU and NPO, where more methods should be included to make the results more solid. Some hyper-parameter sensitivity analysis and statistical report (e.g., std) may also be meaningful.

**Methods And Evaluation Criteria:**

I think the explanations about why IRM works for LLM unlearning should be discussed more. Although the authors have shown some good properties of IRM in Section 5, there remains an open question about why it happens. Forgive me if I am incorrect, it seems that the proposed method tries to find a solution in a relatively smooth loss landscape, or maybe IRM is trying to find a solution in a disentangled parameter space.

Also, the reason why using a single fine-tuning datasets and even the original unlearning dataset can still be useful should be explained more, especially for the latter one. Some analysis and explanations may be required here.

**Other Comments Or Suggestions:**

I raise no other comments.

**Other Strengths And Weaknesses:**

I raise no other comments.

**Questions For Authors:**

Please see the questions above. I think somehow the paper is quite interesting, I will raise my scores if the authors can answer the questions above.

**Relation To Broader Scientific Literature:**

This work enhances the resilience and robustness of LLM unlearning algorithms, which are known to be effective in alleviating or mitigating various negative societal impacts associated with the widespread use of LLMs. These impacts include, but are not limited to, the aspects of avoiding copyright issues and preventing harmful content generation.

**Theoretical Claims:**

Seemingly not applied.

---

> ### Author Rebuttal · Authors · 2025-04-01
>
> Thank you for the reviewer’s review. We respond to the key questions raised below.
> **Q1: Weak motivation for why relearning occurs (in Sec. 4) and under-unlearning in NPO and RMU?**
> **A1:** Thank you for the question. We provide further clarification below. First, we added experiments showing that WGA, is similarly vulnerable to downstream fine-tuning (**[Table R5](https://ibb.co/qMgNgNrm)**). After fine-tuning, both RMU and WGA risk relearning across various datasets. Second, we remark that it may not be fully accurate to categorize NPO and RMU as under-unlearning. For instance, NPO over-unlearns in MUSE ([Table 3, https://arxiv.org/pdf/2407.06460]). And we focus on these methods since they remain SOTAs in the WMDP benchmark.
> Third, the motivation for relearning risk was introduced in Sec. 3 and Fig. 1, where we argue that fine-tuning may conflict with the optimized unlearning direction, potentially reversing its effect. In Sec. 4, we aim to align the unlearning direction with the fine-tuning direction by introducing the concept of IRM, treating fine-tuning as a different training "environment" the unlearned model must adapt to; see Lines 191-219. This also motivates the task vector analysis in Sec. 5 to study the relationship between unlearning and fine-tuning directions.
> **Q2: Definition of the operation ⊥? Is GSM8K completely orthogonal to WMDP? Gradient direction only works under a linear assumption?**
> **A2:** Thank you for your questions. First, the operation ⊥ in Sec. 5 is defined by a cosine value of 0 for the angle between two task vectors (Line 293). We use ⊥ to indicate irrelevance between the fine-tuning task and the unlearning task (Line 249). Second, we did not claim that GSM8K is fully orthogonal to WMDP; rather, it is an irrelevant fine-tuning task for WMDP unlearning. As shown in Fig. 4, the nearly 90-degree angle between the GSM8K vector ($\tau_\mathrm{ft}$) and the WMDP vector ($\tau_\mathrm{ILU}$) demonstrates minimal conflict. Third, our analysis in Sec. 5 is based on the task vector, which reflects the direction from the fine-tuned model to the pre-trained model. We do not claim this as the precise gradient direction (i.e., no linear assumption is imposed). The use of task vectors to investigate task-wise learning direction geometry has also been applied in non-linear cases [R1-3]. Even under a linear assumption, the work in [R2] shows that this is a valid approach for practical task vectors (e.g., [Eq. (3), R2]).
> > [R1]  arXiv:2212.04089
> > [R2]  arXiv:2305.12827
> > [R3]  https://openreview.net/forum?id=vRvVVb0NAz
>
> **Q3: Further explanation on why IRM works for LLM unlearning, a solution in a relatively smooth loss landscape, or a solution in a disentangled parameter space?**
> **A3:** Thank you for this insightful question. We conducted experiments to visualize the forget loss landscape of IRM using the visualization method in https://arxiv.org/pdf/1712.09913, with results presented in  **[Figure R6](https://ibb.co/qLXDgVfJ)**. As shown, applying IRM does not smooth the model's forget loss landscape. This is expected, as the invariance regularization (i.e., the gradient norm term in Eq. (4)) is defined over the fine-tuning datasets rather than the forget data. Its purpose is not to smoothen the unlearning optimization (i.e., the upper-level problem in Eq. (2)), but to ensure that the unlearned model $\boldsymbol \phi$ satisfies the optimality conditions for fine-tuning, i.e., the lower-level constraints in Eq. (2) (Lines 166-169).
> **Q4: Why using a single fine-tuning dataset and even the original unlearning dataset can still be useful should be explained more?.**
> **A4:** Thanks for your question. First, the better performance with a single fine-tuning dataset can be attributed to the IRM-type optimization difficulty in Eq. (2). As shown, using more fine-tuning datasets ($\mathcal D_i$) increases the challenge of achieving a lower-level solution by creating a more restricted constraint space (to satisfy multiple fine-tuning task optimalities simultaneously). This complicates the unlearning optimization in Eq. (4), making it harder to achieve stationarity across all fine-tuning tasks. Second, using the original unlearning dataset (Lines 220–228) remains valuable, especially under the relearning attack[R4], where forget data samples are used for fine-tuning. Still, this may not sufficiently improve robustness for other fine-tuning tasks (Lines 261–270).
> > [R4] arXiv:2406.13356
>
> **Q5: Additional datasets and baselines, and other suggestions.**
> **A5:** First, we added results on the MUSE dataset and compared ILU with TAR; see Q3 and Q2 from [Reviewer b9VT](https://openreview.net/forum?id=x2lm33kdrZ&noteId=aitszercwQ).
> Second, Fig. 6 shows our main hyperparameter sensitivity study. We plan to extend this to the optimizer in the revision and include statistical reporting. Notably, Fig. 5 demonstrates that our robustness performance consistently surpasses the baselines.

---

> > ### Comment · Reviewer_yxbz · 2025-04-02
> >
> > For Q3, it remains unclear why Eq. 2 can lead to both improved unlearning and retention. Also, from my knowledge, IRM works well when the data during training and test at least have some perspectives in common. The reason why IRM can maintain performance on completely orthogonal task is hard to me to understand.
> >
> > For Q4, the authors only mentioned the reason for their design choice yet did not discuss why it can lead to better results. Also from the theoretical analysis of IRM, it try to find some invariant features across datasets. If only one retain dataset is adopted, the model may wrongly select variant features and thereby leading to overfitting.
> >
> > Also, the orthogonal assumption somewhat violates the basic assumption within IRM. So, I think some new theoretical analysis are required considered the completely different goals between original IRM works and you do.

---

> > > ### Author Response · Authors · 2025-04-05
> > >
> > > **Q1: Additional questions on Q3.**
> > >
> > > **Response:** Thank you for raising these valuable questions. Below is further clarification:
> > > **(R1-1)** Both unlearning (upper-level, i.e., forget loss minimization) and retention (lower-level, i.e., fine-tuning loss minimization) are captured by the couped bi-level optimization problem Eq. (2). The lower-level solution, $\mathbf{w}^*(\boldsymbol{\theta})$, guides unlearning conduted at upper level while maintaining optimality for fine-tuning task. Empirically, we find that a single, unrelated fine-tuning task generalizes well for unlearning invariance across tasks, as shown in the task vector geometry analysis in **[Table R1](https://ibb.co/m5fz1V2w)** (extended from Fig. 4). This is likely because other fine-tuning tasks are similarly unrelated to the unlearning task, making their influence comparable. Thus, the invariance captured via one such task is sufficient to guide the bi-level optimization effectively.
> > >
> > > **(R1-2)** At first glance, training and testing data in IRM need to share common elements, such as digit objects in Colored MNIST (Table 2, https://arxiv.org/pdf/2303.02343). However, conventional IRM mainly addresses supervised classification, where training/testing environments may contain spurious correlations between non-core features (e.g., digit color like Red or Blue) and classification labels (e.g., group IDs). These spurious correlations can even be **opposite** across environments. For instance, one environment may associate Red-colored digits with group ID 1, while another flips this, associating group ID 1 with Blue-colored digits. The differing spurious correlations between training and testing environments in conventional IRM make finding invariance more challenging, even when environments share common digits.
> > >
> > > The reviewer seems to infer that the unlearning dataset (WMDP) and fine-tuning dataset (GSM8K) are "orthogonal," making IRM a challenging task. We make the following points:
> > >
> > > a. Based on our earlier discussion of flipped spurious correlations between data features and labels, we believe negative correlations between environments make IRM harder to find invariance, compared to orthogonality, as they introduce conflicts even if the environments share some common aspects.
> > >
> > > b. As explained in (R1-1), invariance optimization in the unlearned model is governed by the coupled bi-level optimization foundation. While the globally optimal solution may not be achievable with orthogonal unlearning and fine-tuning tasks, a good (local) solution is still possible, as shown by our WMDP/MUSE experiments and task vector geometry validation in Fig. 4. While IRM works in its original setting, it can also be extended to unlearning and fine-tuning by following its bi-level optimization foundation.
> > >
> > > **Q2: Additional questions for Q4.**
> > >
> > > **Response:** Apologies for the confusion; it seems our key point wasn’t well delivered in the original response.
> > > **(R2-1)**  Using multiple fine-tuning sets to promote invariance in Eq. (2) **may over-constrain** the lower-level solution space, making it harder to find a good solution and increasing the complexity of the bi-level optimization. To validate this, [Figure R7](https://ibb.co/0p8wdPng) shows that using multiple fine-tuning sets for stationarity regularization (gradient norm regularization in Eq. (4)) makes optimization more difficult, as indicated by the higher loss compared to using a single fine-tuning set over optimization steps.
> > > **(R2-2)** While our motivation stems from IRM for invariance across "multiple environments" (R1-2), the more general rationale lies in its bi-level optimization foundation (R1-1) to align unlearning (upper-level optimization) with fine-tuning optimality (lower-level optimization), reducing conflicts. Even using a single, unrelated retain dataset (e.g., GSM8K), the key is leveraging fine-tuning signals to guide the unlearning-finetuning relationship, minimizing fine-tuning’s negative impact. Instead of the reviewer's overfitting hypothesis, we show that such a dataset has effectively regularized the unlearning process, as also discussed in R1-1.
> > >
> > > **Q3: Orthogonal assumption violates IRM’s basic assumption within IRM.**
> > >
> > > **Response:** As clarified in R1 and R2-2, our proposal extends the original IRM setting while following its bi-level optimization foundation, ensuring unlearning remains robust to fine-tuning’s influence on the unlearned model. The orthogonality assumption applies between the optimized unlearning and fine-tuning tasks (see Line 218, Sec. 5, and Fig. 4), not between training and test-time fine-tuning environments. We feel this orthogonality does not violate IRM's basic assumption, as explained in R1-2.  As shown in  **[Table R1](https://ibb.co/m5fz1V2w)**, the ILU-derived unlearning task vector indeed remains nearly orthogonal to different test-time fine-tuning task vectors, which differ from the GSM8K task used during training.

---

### Official Review · Reviewer_b9VT · 2025-03-13

**Overall Recommendation:** 3

**Summary:**

The authors propose a novel method to enhance the robustness of language model unlearning against fine-tuning. The core contribution is the introduction of invariance regularization, inspired by Invariant Risk Minimization, which aims to make unlearning effects resilient to subsequent fine-tuning. The paper demonstrates that existing unlearning methods often lose their efficacy after even minimal fine-tuning, and provides both theoretical analysis and empirical evidence that the proposed regularization significantly improves robustness.

**Claims And Evidence:**

Yes

**Essential References Not Discussed:**

To best of my knowledge, there are no essential related works missed.

**Experimental Designs Or Analyses:**

Yes, I checked. There are no issues.

**Methods And Evaluation Criteria:**

The evaluation focuses exclusively on benign fine-tuning with clean data. However, in adversarial settings, more aggressive approaches (like targeted relearning attacks) might be employed to reverse unlearning effects. The paper would be stronger if it evaluated against such stronger attack scenarios to demonstrate the limits and capabilities of the proposed method.

**Other Comments Or Suggestions:**

No.

**Other Strengths And Weaknesses:**

Strengths:

(1) The paper addresses an important but under-explored aspect of machine unlearning - maintaining unlearning effects after model fine-tuning. As language models continue to be deployed, updated, and fine-tuned in practice, ensuring the persistence of unlearning is a critical safety concern. The framing of this problem is novel and timely.
(2) The task vector analysis provided to explain why the method works is effective and insightful. By connecting the geometry of task vectors between unlearning and fine-tuning, the authors provide substantive evidence for why invariance regularization prevents the "forgetting of forgetting" phenomenon. This analysis helps clarify the underlying mechanism of the proposed solution.
(3) The paper is well-written and logically structured. The authors clearly articulate the problem, motivate their approach, and present results in an accessible manner. The figures effectively illustrate the key concepts and findings, making the technical content more approachable.

Weaknesses:

(1) My primary concern is the absence of comparisons with existing methods designed to enhance unlearning robustness. Particularly, works like "Tamper-resistant safeguards" have been proposed specifically to make unlearning methods more robust. Without direct comparisons to such baselines, it is difficult to assess the relative advantages of the proposed approach. This omission significantly limits the strength of the empirical claims.
(2) The evaluation focuses exclusively on benign fine-tuning with clean data. However, in adversarial settings, more aggressive approaches (like targeted relearning attacks) might be employed to reverse unlearning effects. The paper would be stronger if it evaluated against such stronger attack scenarios to demonstrate the limits and capabilities of the proposed method.
(3) The empirical evaluation is conducted on a single language model architecture. Given the significant architectural differences among modern LLMs (Llama, Qwen, etc.), it would strengthen the paper to demonstrate that the benefits of invariance regularization generalize across different model families and sizes.

**Questions For Authors:**

I will change my evaluation if the following questions could be solved:

(1) Could you provide the comparison and analysis of other baselines like "Tamper-Resistant Safeguards for Open-Weight LLMs".
(2) The paper would be stronger if it evaluated against stronger attack scenarios to demonstrate the limits and capabilities of the proposed method.
(3) If experiments could be conducted on more LLMs, it would help demonstrate the scalability of the proposed method.

**Relation To Broader Scientific Literature:**

The paper addresses an important but under-explored aspect of machine unlearning - maintaining unlearning effects after model fine-tuning.

**Theoretical Claims:**

Yes, I checked. There are no issues.

---

> ### Author Rebuttal · Authors · 2025-04-01
>
> **Q1: Robust unlearning evaluation on aggressive approaches (like targeted relearning attacks).**
>
> **A1:** We appreciate the reviewer’s suggestion and have incorporated additional experiments to evaluate our method under the **relearning attack** setting [R1]. Specifically, the attack involves fine-tuning the unlearned model for one epoch using a small subset of **forget samples** (e.g., 60 samples in our experiments). **[Table R2](https://ibb.co/DHd5Zw2c)** shows the forget quality (FQ) metric before and after the relearning attack for **ILU** methods, compared to **NPO**, **RMU**, and **TAR** (which uses meta-learning for robustness against harmful fine-tuning) [R2]. Note that TAR was implemented on the LLaMA3-8B model, and we maintained a consistent experimental setup.  As shown in **[Table R2](https://ibb.co/DHd5Zw2c)**, ILU-based methods demonstrate significantly stronger robustness against the relearning attack compared to the baseline NPO and RMU, as evidenced by the much smaller FQ drop after the attack. The performance of the new baseline TAR is also included. As seen, ILU and TAR exhibit nearly identical robustness, with only a small FQ drop of 0.04–0.05. However, as discussed in Lines 55–61, we did not use TAR due to its significantly high computational cost. We defer the discussion of TAR's computational limitations to the next question.
> > [R1] Hu, Shengyuan, et al. "Unlearning or Obfuscating? Jogging the Memory of Unlearned LLMs via Benign Relearning." to appear in ICLR’25.
> > [R2] Tamirisa, Rishub, et al. "Tamper-resistant safeguards for open-weight llms." arXiv preprint arXiv:2408.00761 (2024).
>
> **Q2: Comparison with "Tamper-Resistant Safeguards for Open-Weight LLMs".**
>
> **A2:** Thank you for the suggestion. In the rebuttal, we included **TAR** as an additional baseline, which is based on the NPO loss, uses **LLaMA3-8B-Instruct** as the base model, and achieves model robustness through a **meta-learning** approach. To ensure a fair comparison, we also developed NPO+ILU(GSM8K) on the same LLaMA3-8B-Instruct model and compared their performances. As shown in **[Table R3](https://ibb.co/4ZSkRQGh)**, both TAR and our proposed ILU achieve similar unlearning performance before and after downstream fine-tuning. For the average robust accuracy (RA), ILU decreases RA by 0.03 post fine-tuning, slightly more than TAR’s decrease of 0.02. Consistent robustness performance was also observed in **[Table R2](https://ibb.co/DHd5Zw2c)** against the relearning attack. However, TAR introduces significantly higher computational overhead compared to ILU. **[Table R3](https://ibb.co/4ZSkRQGh)** also measures the total training time for both methods. We observe that TAR, being based on meta-learning, incurs much higher time consumption due to the multi-step gradient unrolling required for robustness against fine-tuning. Specifically, TAR takes 7,441.9 minutes, while ILU only requires 118.2 minutes, achieving the same performance and being **63 times faster than TAR**.  The above results highlight that ILU not only achieves highly competitive unlearning robustness but also significantly improves efficiency, making it a more practical choice for LLM unlearning.
>
> **Q3: Additional experiments on different LLMs.**
>
> **A3:** We understand the reviewer’s concern regarding the **model choice** and the **consistent effectiveness** of our proposal. To address this, we demonstrate the performance of **ILU** on the larger **LLaMA3-8B-Instruct model**, comparing it with **TAR** (**[Table R3](https://ibb.co/4ZSkRQGh)** and **[Table R2](https://ibb.co/DHd5Zw2c)**).  Additionally, we extended our experiments to the **MUSE LLM unlearning benchmark** (https://arxiv.org/abs/2407.06460), which includes: (a) **MUSE-News**, based on BBC News, using the **LLaMA-2 7B** model as the target, and (b) **MUSE-Books**, based on Harry Potter books, using the **ICLM-7B** model as the target. **[Table R4](https://ibb.co/m5Rtqc3t)** compares the performance of ILU with NPO, where forget quality (FQ) is measured by KnowMem and VerbMem on $D_f$, along with fine-tune accuracy (FA) after fine-tuning. As seen, before fine-tuning, both NPO and NPO+ILU (GSM8K) demonstrate strong unlearning effectiveness across MUSE-News and MUSE-Books. After fine-tuning, the unlearning performance (KnowMem and VerbMem on $D_f$) is largely preserved for ILU (staying at low values), while NPO shows a clear increase in these metrics post-fine-tuning.

---

### Official Review · Reviewer_f9uQ · 2025-03-13

**Overall Recommendation:** 4

**Summary:**

This paper addresses the challenge of machine unlearning in large language models (LLMs) by improving the robustness of removing targeted knowledge while preserving model utility. Existing unlearning methods are highly sensitive to downstream fine-tuning, often leading to the unintended recovery of unlearned information, even when the fine-tuning task is unrelated. To address this, the authors introduce invariance regularization inspired by Invariant Risk Minimization (IRM) and propose a new framework called Invariant LLM Unlearning (ILU), which enhances resistance to fine-tuning-induced recovery. Extensive experiments on the WMDP benchmark show that ILU significantly outperforms state-of-the-art unlearning methods, such as negative preference optimization (NPO) and representation misdirection for unlearning (RMU).

**Claims And Evidence:**

The paper claims that unlearned concepts can be recovered by finetuning on an irrelevant dataset. The claim is supported by the prior work as well as experiments shown in this work. The paper proposes an invariant unlearning approach built on invariant risk minimization. The performance of the proposed method is supported with experiments in the paper. Overall, I think the claims in this work are well-supported by experiments and prior research.

**Essential References Not Discussed:**

The paper has properly cited the related works.

**Experimental Designs Or Analyses:**

Experiment on six finetuning datasets and WMDP is appropriate.

**Methods And Evaluation Criteria:**

The proposed unlearning method based on IRM is sound. Evaluation criteria on unlearning accuracy and finetuning accuracy make sense.

**Other Comments Or Suggestions:**

The task vector results are somewhat confusing. The difference seems not significant (0.16 vs 0.09). I am not sure if the task vector analysis is averaged or just for a single sample. The author should explain the significance of the improved cosine metric as well as incorporate more data samples & tasks to ensure the generality of this study.

**Other Strengths And Weaknesses:**

As mentioned above.

**Questions For Authors:**

See above.

**Relation To Broader Scientific Literature:**

This work leverages the finding in prior literature that unlearning is sensitive to finetuning and adversarial recovery. The proposed method is based on IRM.

**Theoretical Claims:**

This work does not have any theoretical claim.

---

> ### Author Rebuttal · Authors · 2025-04-01
>
> We thank the reviewer for the detailed summary of our work and contributions. We also appreciate the insightful comments and provide our detailed responses below.
>
> **Q1:  Confusion in Fig. 4 task vector.**
> **A1:** We apologize for any confusion caused in Fig. 4 and our presentation. We will improve them in the revision.  The difference between the two angles (or cosine values) shown in Fig. 4 is not intended to explain the robustness gain of ILU over NPO. Instead, these angles serve to validate our geometry plots and are later used in our analysis (between Lines 318-Left and Line 285-Right in our submission). In summary, the sign flip (Line 281-Right)  when comparing $ \cos(\angle (\tau_{\text{NPO} \to \text{ft}}, \tau_{\text{NPO}})) = -0.41 < 0$ and $ \cos(\angle (\tau_{\text{ILU} \to \text{ft}}, \tau_{\text{ILU}})) = 0.09 > 0$ demonstrates that NPO post-fine-tuning results in a much larger deviation from its original unlearning direction (an obtuse angle), whereas ILU remains near orthogonal, better disentangling the fine-tuning effect from the original unlearning and preserving the unlearning direction within the unlearning space after fine-tuning, as shown by our geometric validation in Fig. 4.
>
> **Q2: On the significance of task vector analysis for more samples/tasks.**
> **A2:** The task vector is defined as the difference between the fine-tuned/unlearned model and its base model over the fine-tuning dataset, as illustrated in Lines 278-287. Therefore, it depends on the entire fine-tuning task rather than a single sample. To better highlight the significance of our analysis, we follow the reviewer’s suggestion to conduct a task vector analysis across additional fine-tuning tasks within the NPO-based unlearning context. See results in **[Table R1](https://ibb.co/m5fz1V2w)**, which reported two metrics: $ \cos(\angle (\tau_{\text{NPO} \to \text{ft}}, \tau_{\text{NPO}}))$ and $ \cos(\angle (\tau_{\text{ILU} \to \text{ft}}, \tau_{\text{ILU}}))$, as clarified earlier in Fig. 4. A smaller negative value for the former $ \cos(\angle (\tau_{\text{NPO} \to \text{ft}}, \tau_{\text{NPO}}))$ indicates that the fine-tuning task vector forms a larger obtuse angle with the unlearning task vector, implying greater conflict between the two. In contrast, the cosine value closer to 0 for the latter $ \cos(\angle (\tau_{\text{ILU} \to \text{ft}}, \tau_{\text{ILU}}))$ demonstrates the effectiveness of our method, as it has less conflict between the fine-tuning and unlearning directions.

---

### Decision · Program_Chairs · 2025-05-01

**Decision:**

Accept (poster)

**Comment:**

Existing unlearning methods are highly sensitive to downstream fine-tuning, often leading to the unintended recovery of unlearned information, even when the fine-tuning task is unrelated. To enhance resistance to such fine-tuning-induced recovery, this paper proposes a new machine unlearning framework called Invariant LLM Unlearning (ILU), which is based on invariance regularization and inspired by Invariant Risk Minimization (IRM).

The reviewers appreciate the following strengths: 1) This paper addresses an important but under-explored aspect of machine unlearning—maintaining unlearning effects after model fine-tuning; 2) This paper provides task vector analysis to interpret the underlying mechanism of the proposed solution; 3) This paper is well-written and well-organized.

The reviewers also point out the following weaknesses: 1) Lacks certain empirical results, such as comparisons with existing methods designed to enhance unlearning robustness, fine-tuning in the adversarial setting, diverse LLM architectures, and more datasets beyond TOFU, MUSE, and the Harry Potter dataset; 2) Lacks sufficient clarification of the relearning scenario and generalizes the verification of the relearning issue to GA-based unlearning algorithms; 3) Lacks sufficient explanation of the effectiveness of IRM and the utilization of a single fine-tuning dataset; 4) Contains several flaws in experimental design.

During the discussion, the authors further provided: 1) New experimental results on a larger LLM architecture; 2) Comparisons with additional baselines; 3) Detailed clarification and explanation regarding the relearning scenario, the effectiveness of IRM, and the utilization of a single fine-tuning dataset; 4) Clarifications regarding flaws in experimental design.

Three of the reviewers are positive after the discussion, and one reviewer still remains negative about the paper. The paper is at the borderline for acceptance. Therefore, the area chair recommends weak acceptance of the paper, with the final decision left to the SAC and PCs.